# HIGH-CAPACITY EXPERT BINARY NETWORKS

**Adrian Bulat**
Samsung AI Cambridge
adrian@adrianbulat.com

**Brais Martinez**
Samsung AI Cambridge
brais.a@samsung.com

**Georgios Tzimiropoulos**
Samsung AI Cambridge
Queen Mary University of London, UK
g.tzimiropoulos@qmul.ac.uk

## ABSTRACT

Network binarization is a promising hardware-aware direction for creating efficient deep models. Despite its memory and computational advantages, reducing the accuracy gap between binary models and their real-valued counterparts remains an unsolved challenging research problem. To this end, we make the following 3 contributions: (a) To increase *model capacity*, we propose Expert Binary Convolution, which, for the first time, tailors conditional computing to binary networks by learning to select one data-specific expert binary filter at a time conditioned on input features. (b) To increase *representation capacity*, we propose to address the inherent information bottleneck in binary networks by introducing an efficient width expansion mechanism which keeps the binary operations within the same budget. (c) To improve *network design*, we propose a principled binary network search mechanism that unveils a set of network topologies of favorable properties. Overall, our method improves upon prior work, *with no increase in computational cost*, by $\sim 6\%$, reaching a groundbreaking $\sim 71\%$ on ImageNet classification. Code will be made available here.

## 1 INTRODUCTION

A promising, hardware-aware, direction for designing efficient deep learning models case is that of network binarization, in which filter and activation values are restricted to two states only: $\pm 1$ (Rastegari et al., 2016; Courbariaux et al., 2016). This comes with two important advantages: (a) it compresses the weights by a factor of $32\times$ via bit-packing, and (b) it replaces the computationally expensive multiply-add with bit-wise `xnor` and `popcount` operations, offering, in practice, a speed-up of $\sim 58\times$ on a CPU (Rastegari et al., 2016). Despite this, how to reduce the accuracy gap between a binary model and its real-valued counterpart remains an open problem and it is currently the major impediment for their wide-scale adoption.

In this work, we propose to approach this challenging problem from 3 key perspectives:
**1. Model capacity:** To increase model capacity, we firstly introduce the first application of Conditional Computing (Bengio et al., 2013; 2015; Yang et al., 2019) to the case of a binary networks, which we call Expert Binary Convolution. For each convolutional layer, rather than learning a weight tensor that is expected to generalize well across the entire input space, we learn a set of $N$ experts each of which is tuned to specialize to portions of it. During inference, a very light-weight gating function dynamically selects a single expert for each input sample and uses it to process the input features. Learning to select a *single*, tuned to the input data, expert is a key property of our method which renders it suitable for the case of binary networks, and contrasts our approach to previous works in conditional computing (Yang et al., 2019).
**2. Representation capacity:** There is an inherent information bottleneck in binary networks as only 2 states are used to characterize each feature, which hinders the learning of highly accurate models. To this end, for the first time, we highlight the question of depth vs width in binary networks and propose a surprisingly unexplored efficient mechanism for increasing the effective width of the network by preserving the original computational budget. We show that our approach leads

to noticeable gains in accuracy without increasing computation.

**3. Network design:** Finally, and inspired by similar work in real-valued networks (Tan & Le, 2019), we propose a principled approach to search for optimal directions for scaling-up binary networks.

**Main results:** Without increasing the computational budget of previous works, our method improves upon the state-of-the-art (Martinez et al., 2020) by $\sim 6\%$, reaching a groundbreaking $\sim 71\%$ on ImageNet classification.

## 2  RELATED WORK

### 2.1  NETWORK BINARIZATION

Since the seminal works of Courbariaux et al. (2015; 2016) which showed that training fully binary models (both weights and activations) is possible, and Rastegari et al. (2016) which reported the very first binary model of high accuracy, there has been a great research effort to develop binary models that are competitive in terms of accuracy when compared to their real-valued counterparts, see for example (Lin et al., 2017; Liu et al., 2018; Alizadeh et al., 2018; Bulat et al., 2019; Bulat & Tzimiropoulos, 2019; Ding et al., 2019; Wang et al., 2019; Zhuang et al., 2019; Zhu et al., 2019; Kim et al., 2020; Bulat et al., 2020; Martinez et al., 2020). Notably, many of these improvements including real-valued down-sampling layers (Liu et al., 2018), double skip connections (Liu et al., 2018), learning the scale factors (Bulat & Tzimiropoulos, 2019), PReLUs (Bulat et al., 2019) and two-stage optimization (Bulat et al., 2019) have been put together to build a strong baseline in Martinez et al. (2020) which, further boosted by a sophisticated distillation and data-driven channel rescaling mechanism, yielded an accuracy of $\sim 65\%$ on ImageNet. This method, along with the recent binary NAS of Bulat et al. (2020) reporting accuracy of $\sim 66\%$, are to our knowledge, the state-of-the-art in binary networks.

Our method further improves upon these works achieving an accuracy of $\sim 71\%$ on ImageNet, crucially without increasing the computational complexity. To achieve this, to our knowledge, we propose for the first time to explore ideas from Conditional Computing (Bengio et al., 2013; 2015) and learn data-specific binary expert weights which are dynamically selected during inference conditioned on the input data. Secondly, we are the first to identify width as an important factor for increasing the representation capacity of binary networks, and introduce a surprisingly simple yet effective mechanism to enhance it without increasing complexity. Finally, although binary architecture design via NAS (Liu et al., 2018; Real et al., 2019) has been recently explored in (Kim et al., 2020; Bulat et al., 2020), we propose to approach it from a different perspective that is more related to Tan & Le (2019), which was developed for real-valued networks.

### 2.2  CONDITIONAL COMPUTATION

Conditional computation is a very general data processing framework which refers to using different models or different parts of a model conditioned on the input data. Wang et al. (2018) and Wu et al. (2018) propose to completely bypass certain parts of the network during inference using skip connections by training a policy network via reinforcement learning. Gross et al. (2017) proposes to train large models by using a mixture of experts trained independently on different partitions of the data. While speeding-up training, this approach is not end-to-end trainable nor tuned towards improving the model accuracy. Shazeer et al. (2017) trains thousands of experts that are combined using a noisy top-k expert selection while Teja Mullapudi et al. (2018) introduces the HydraNet in which a routing function selects and combines a subset of different operations. The later is more closely related to online network search. Chen et al. (2019) uses a separate network to dynamically select a variable set of filters while Dai et al. (2017) learns a dynamically computed offset.

More closely related to the proposed `EBConv` is Conditional Convolution, where Yang et al. (2019) propose to learn a Mixture of Experts, i.e. a set of filters that are linearly combined using a routing function. In contrast, our approach learns to select a *single* expert at a time. This is critical for binary networks for two reasons: (1) The linear combination of a binary set of weights is non-binary and, hence, a second binarization is required giving rise to training instability and increased memory consumption. In Section 5, we compare with such a model and show that our approach works significantly better. (2) The additional computation to multiply and sum the weights, while negligible for real-valued networks, can lead to a noticeable computational increase for binary ones.

Finally, we note that our single expert selection mechanism is akin to the Gumbel-max trick (Gumbel, 1948) and the Gumbel-Softmax Estimator (Jang et al., 2016; Maddison et al., 2016) previously used in various forms for NAS (Chang et al., 2019), multi-task learning (Guo et al., 2020) and variational auto-encoders (Jang et al., 2016). To our knowledge, the proposed `EBConv` is the very first adaptation for conditional computing within binary neural networks.

## 3 BACKGROUND ON BINARY NETWORKS

Following Rastegari et al. (2016); Bulat & Tzimiropoulos (2019), a binary convolution is defined as:

$$\texttt{BConv}(\mathbf{x}, \boldsymbol{\theta}) = (\texttt{sign}(\mathbf{x}) \circledast \texttt{sign}(\boldsymbol{\theta})) \odot \alpha, \tag{1}$$

where $\mathbf{x}$ is the input, $\boldsymbol{\theta}$ the weights, $\circledast$ denotes the binary convolutional operation, $\odot$ the Hadamard product, and $\alpha \in \mathbb{R}^C$ is learned via back-propagation, as in Bulat & Tzimiropoulos (2019).

The binarization is performed in two stages Bulat et al. (2019); Martinez et al. (2020). During Stage I, we train a network with binary activations and real-valued weights. Note that the accuracy of Stage I models are very representative to that of the final fully binary model (see Table 4). During Stage II, we initialize from Stage I to train a network with both weights and activations binary. When reporting results, if no stage is specified, the model (weights and activations) is fully binary.

We set as baseline the *Strong Baseline* model (denoted as *SBaseline*) from Martinez et al. (2020) on top of which we implemented the proposed method. We denote as *Real-to-bin* their full model.

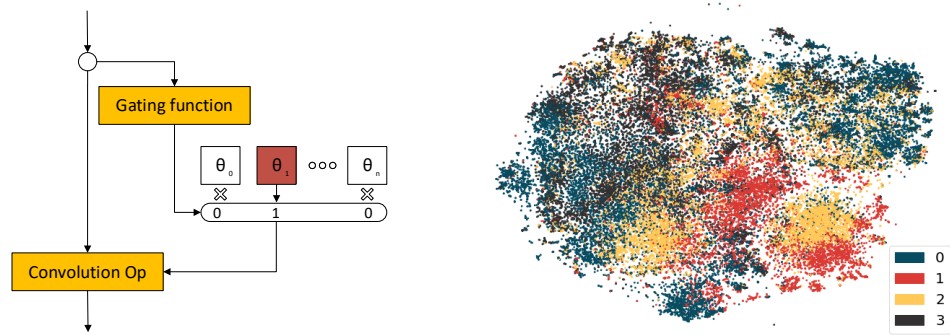

(a) The proposed Expert Binary Convolution (`EBConv`). Note that only a *single* expert is active at a time.

(b) 2D t-SNE embeddings of ImageNet validation set for a model with 4 experts and Top-1 acc. of 63.8%.

Figure 1: (a) Schematic representation of the proposed `EBConv` layer, and (b) t-SNE embedding visualisation of the features before the classifier along with the corresponding expert that was activated for each sample. Points located closer to each other are more semantically and visually similar. Each data point is coloured according to the expert selected by the last EBConv from our network. As multiple clusters emerge in the figure, it can be deduced that the experts learned a preference for certain classes, or groups of classes from ImageNet. This suggests that the EBConv layer learned semantically meaningful representations of the data. Best viewed in color.

## 4 METHOD

### 4.1 EXPERT BINARY CONVOLUTION

Assume a binary convolutional layer with input $\mathbf{x} \in \mathbb{R}^{C_{in} \times W \times H}$ and weight tensor $\boldsymbol{\theta} \in \mathbb{R}^{C_{in} \times C_{out} \times k_H \times k_W}$. In contrast to a normal convolution that applies the same weights to all input features, we propose to learn a set of expert weights (or simply experts) $\{\boldsymbol{\theta}_0, \boldsymbol{\theta}_1, ..., \boldsymbol{\theta}_{N-1}\}$, $\boldsymbol{\theta}_i \in \mathbb{R}^{C_{in} \times C_{out} \times k_H \times k_W}$ alongside a selector gating function which, given input $\mathbf{x}$, selects *only a single expert* to be applied to it. The proposed `EBConv` layer is depicted in Fig. 1a. To learn the experts, let us first stack them in matrix $\boldsymbol{\Theta} \in \mathbb{R}^{N \times C_{in} C_{out} k_H k_W}$. We propose to learn the following

function:
$$\texttt{EBConv}(\mathbf{x}, \boldsymbol{\theta}) = \texttt{BConv}\left(\mathbf{x}, \left(\varphi(\psi(\mathbf{x}))^T \boldsymbol{\Theta}\right)_r\right), \tag{2}$$

where $\varphi(.)$ is a gating function (returning an $N-$dimensional vector as explained below) that implements the expert selection mechanism using as input $\psi(\mathbf{x})$ which is an aggregation function of the input tensor $\mathbf{x}$, and $(.)_r$ simply reshapes its argument to a tensor of appropriate dimensions.

**Gating function $\varphi$:** A crucial component of the proposed approach is the gating function that implements the expert selection mechanism. An obvious solution would be to use a Winners-Take-All (`WTA`) function, however this is not differentiable. A candidate that comes in mind to solve this problem is the softargmax with temperature $\tau$: as $\tau \to 0$, the entry corresponding to the max will tend to 1 while the rest to 0. However, as $\tau \to 0$, the derivative of the softargmax converges to the Dirac function $\delta$ which provides poor gradients and hence hinders the training process. This could be mitigated if a high $\tau$ is used, however this would require hard thresholding at test time which, for the case of binary networks, and given that the models are trained using Eq. 2, leads to large errors.

To mitigate the above, and distancing from reinforcement learning techniques often deployed when discrete decisions need to be made, we propose, for the forward pass, to use a `WTA` function for defining $\varphi(.)$, as follows:
$$\varphi(z) = \begin{cases} 1, & \text{if } i = \texttt{argmax}(z) \\ 0, & \text{otherwise} \end{cases}. \tag{3}$$

Note that we define $\varphi$ as $\varphi : \mathbb{R}^C \to \mathbb{R}^N$ i.e. as a function that returns an $N-$dimensional vector which is used to multiply (element-wise) $\boldsymbol{\Theta}$ in Eq. 2. This is crucial as, during training, we wish to back-propagate gradients for the non-selected experts. To this end, we propose, for the backward pass, to use the `Softmax` function for approximating the gradients $\varphi(.)$:
$$\frac{\partial \phi}{\partial z} := \frac{\partial}{\partial z}\text{Softmax}(z). \tag{4}$$

Overall, our proposal, `WTA` for forward and `Softmax` for backward, effectively addresses the mismatch during inference between training and testing while, at the same time, it allows meaningful gradients to flow to all experts during training. In Section A.3.3 of the appendix, we also explore the impact of adding a temperature to the softmax showing how its value affects the training process. Note that backpropagating gradients for the non-selected experts applies to the gating function, only; the binary activations and weights continue to use the STE introduced in (Courbariaux et al., 2016; Rastegari et al., 2016).

**Aggregation function $\psi$:** The purpose of this function is to give a summary of the input feature tensor which will be used to select the expert. To avoid overfitting and to keep the computational cost low, we opt for a simple and fast linear function:
$$\psi(\mathbf{x}) = \left[\bar{\mathbf{x}}^{[0]}\bar{\mathbf{x}}^{[1]}\cdots\bar{\mathbf{x}}^{[C-1]}\right]\boldsymbol{\omega}, \tag{5}$$

where $\bar{\mathbf{x}}^{[i]} = \frac{1}{HW}\mathbf{x}^{[i]}$ is the spatial average of the $i-$th channel and $\boldsymbol{\omega} \in \mathbb{R}^{C \times N}$ a learnable projection matrix. Note that no other non-linearity was used as the `WTA` function is already a non-linear function.

**Data-specific experts:** One expected property of `EBConv` implied by the proposed design is that the experts should specialize on portions of data. This is because, for each data sample, a single expert is chosen per convolutional layer. Fig. 1b confirms this experimentally by t-SNE embedding visualisation of the features before the classifier along with the corresponding expert that was activated for each sample of the ImageNet *validation set*.

**Optimization policy:** As in Bulat et al. (2019), we adopt a two-stage training policy where firstly the input features are binarized while learning real-valued weights, and then both input and weights are binarized.

Table 1: Comparison on ImageNet for different number of experts. **All models have the same number of BOPs, including Martinez et al. (2020).**

| #experts | Accuracy (%) | |
|---|---|---|
| | Top-1 | Top-5 |
| 1 (SBaseline) (Martinez et al., 2020) | 60.9 | 83.0 |
| 4 | 63.8 | 85.1 |
| 8 | 64.0 | 85.3 |

Note that the aggregation function $\psi$ is kept real across
all steps since its computational cost is insignificant. Furthermore, due to the discrete decision making process early on, the training can be unstable. Therefore, to stabilize the training we firstly train one expert, and then use this to initialize the training of all $N$ experts. This ensures that early on in the process any decision made by the gating function is a good decision. Overall, our optimization policy can be summarized as follows:

1. Train one expert, parametrized by $\boldsymbol{\theta}_0$, using real weights and binary activations.
2. Replicate $\boldsymbol{\theta}_0$ to all $\boldsymbol{\theta}_i, i = \{1, N-1\}$ to initialize matrix $\boldsymbol{\Theta}$.
3. Train the model initialized in step 2 using real weights and binary activations.
4. Train the model obtained from step 3 using binary weights and activations.

## 4.2 ENHANCING BINARY INFORMATION FLOW

While the previous section addressed the issue of model capacity, in this section, we address the problem of the representation capacity of the binary activations. This issue arises due to the fact that only 2 states are used to characterize each feature, resulting in an information bottleneck which hinders the learning of highly accurate binary networks. To our knowledge, there is little prior work which explicitly tries to solve this problem (Liu et al., 2018).

Our solution is surprisingly simple yet effective: the only parameters one can adjust in order to increase the representational power of binary features are the resolution and the width (i.e. number of channels). The former is largely conditioned on the resolution of the data, being as such problem dependent. Hence, we propose the latter, which is to increase the network width. For example a width expansion of $k = 2$ can increase the number of unique configurations for a $32 \times 7 \times 7$ binary feature tensor from $2^{32 \times 7 \times 7} = 2^{1568}$ to $2^{2^{136}}$. However, increasing the network width directly causes a quadratic increase in complexity with respect to $k$. Hence, in order to keep the number of binary operations (BOPs) constant, we propose to use *Grouped Convolutions* with group size $G$ pro-

Table 2: Comparison on ImageNet for different number of experts and expansion rates. **All models have the same number of BOPs, including Martinez et al. (2020).**

| Expansion | # experts | Accuracy (%) | |
| --- | --- | --- | --- |
| | | Top-1 | Top-5 |
| 1 (SBaseline) (Martinez et al., 2020) | 1 | 60.9 | 83.0 |
| 2 | 1 | 64.6 | 85.6 |
| 4 | 1 | 65.1 | 86.0 |
| 1 | 4 | 63.8 | 85.1 |
| 1 | 8 | 64.0 | 85.3 |
| 2 | 4 | 66.0 | 86.4 |
| 2 | 8 | 66.3 | 86.6 |

portional to the width expansion, i.e. $G = k^2$. Note that we do not simply propose using grouped convolutions within binary networks as in (Phan et al., 2020; Bulat et al., 2020). We propose width expansion to address the inherent information bottleneck within binary networks and use grouped convolutions as a mechanism for increasing the capacity while preserving the computational budget fixed. Moreover, we note that since we are using grouped convolutions, features across groups need to be somehow combined throughout the network. This can be achieved at no extra cost through the $1 \times 1$ convolutions used for downsampling at the end of each stage where change of spatial resolution occurs. In Section 4.3, we further propose a more effective way to achieve this, based on binary $1 \times 1$ convolutions, which however add some extra complexity. Moreover, in Section 4.3, we will further propose to search for the optimal group size depending on the layer location.

As Table 2 clearly shows, models trained with a width multiplier higher than 1 offer consistent accuracy gains, notably without increasing complexity. Importantly, these gains also add up with the ones obtained by using the proposed `EBConv`. This is not surprising as width expansion improves representation capacity while the expert increases model capacity.

## 4.3 DESIGNING BINARY NETWORKS

In general, there is little work in network design for binary networks. Recently, a few binary NAS techniques have been proposed (Kim et al., 2020; Shen et al., 2019; Bulat et al., 2020). Despite reporting good performance, these methods have the same limitations typical of NAS methods, for

example, having to search for an optimal cell using a predefined network architecture, or having to hand pick the search space. Herein, and inspired by Tan & Le (2019), we propose a mixed semi-automated approach that draws from the advantages of both automatic and manual network designing techniques. Specifically, setting the standard ResNet-18 (He et al., 2016) network as a starting point, we focus on searching for optimal binary network structures, gradually exploring a set of different directions (width, depth, groups, layer arrangement).

**Effect of block arrangement:** Starting from a ResNet-based topology in mind, we denote a network with $N_i, i = \{1, 2, 3, 4\}$ blocks at each resolution as $N_0 N_1 N_2 N_3$, with each block having two convolutional layers. We first investigate if re-arranging the blocks, mainly by using a network which is heavier at later stages, can have an impact on accuracy. Note that since the number of features is doubled among stages, this re-arrangement preserves the same complexity. Table 3 shows the results. As it can be observed the accuracy remains largely unchanged while the layers are re-distributed.

**Depth vs width:** In Section 4.2, we proposed an efficient width expansion mechanism which is found to increase the accuracy of binary networks without increasing complexity. Herein, we evaluate the effect of increasing depth by adding more blocks. Fig. 2a shows the results of depth expansion. Each constellation represents a different architecture out of which we vary only the number of blocks, i.e. the depth. As we may clearly see, the returns of increasing depth are diminished as complexity also rapidly increases, resulting in very heavy models. Note that previous work for the case of real-valued

Table 3: Comparison on ImageNet between stage I models with different block arrangements.

| $N_0 N_1 N_2 N_3$ | Accuracy (%) | |
| --- | --- | --- |
| | **Top-1** | **Top-5** |
| 1133 | 63.8 | 86.5 |
| 1142 | 63.8 | 86.8 |
| 1124 | 63.7 | 87.4 |
| 2222 | 63.9 | 87.4 |

networks (Zagoruyko & Komodakis, 2016) has shown that wide models can perform as well as deep ones. Our results show that, for a fixed computation budget, the proposed wide binary models with grouped convolutions actually outperform the deep ones by a large margin.

**Effect of aggregation over groups:** Our efficient width expansion mechanism of Section 4.2 uses a very weak way of aggregating the information across different groups. A better way is to explicitly use a $1 \times 1$ binary convolutional layer (with no groups) after each block. The effect of adding that layer is shown in Fig. 3. Clearly, aggregation across groups via $1 \times 1$ convolutions offers significant accuracy gains, adding at the same time a reasonable amount of complexity.

**Effect of groups:** In Section 4.2, we proposed grouped convolutions as a mechanism for keeping the computations under control as we increase the network width. Herein, we go one step further and explore the effect of different group sizes and their placement across the network. This, in turn, allows, with a high degree of granularity, to vary the computational budget at various points in the network while preserving the width and as such the information flow. To describe the space of network structures explored, we use the following naming convention: we denote a network with $N_i, i = \{1, 2, 3, 4\}$ blocks at each resolution, a corresponding width expansion $E$ (the same $E$ was used for all blocks) and group size $G_i$ for each convolution in these blocks as: $N_0 N_1 N_2 N_3 - E - G_0 : G_1 : G_2 : G_3$.

As the results from Fig. 2b and Table 4 show, increasing the number of groups (especially for the last 2 stages) results in significantly more efficient models which maintain the high accuracy (with only small decrease) compared to much larger models having the same network structure but fewer groups. Our results suggest that group sizes of 16 or 32 for the last 2 stages provide the best trade-off.

**Network search strategy:** In summary, the network search space used in this work consists of the following degrees of freedom: a) rearranging the blocks, b) defining the depth of the model, c) defining the width at each stage, and finally d) selecting the optimal number of groups per each stage. In order to search for the optimal configuration, we gradually search in each direction separately while keeping all the others fixed. Then, we identify a set of promising search directions which we then combine to train new candidates. We repeat this step one more time and, then, from the final population of candidates we select the best models shown in Table 4. This procedure results in models that outperform recently proposed binary NAS methods (Bulat et al., 2020; Kim et al., 2020) by more than $5\%$ while also being more computationally efficient.

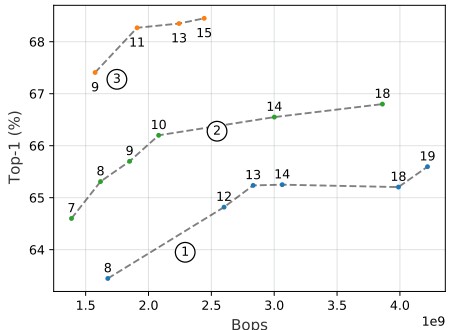 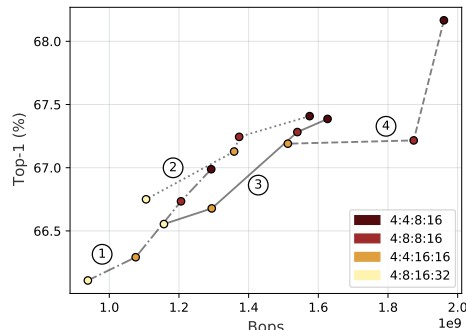

(a) Effect of depth on accuracy. Each constellation represents a different network from which we vary only the number of blocks (shown by the annotated text), i.e. the depth. Increasing depth has diminishing returns. The specific networks for each constellation are described in Table 6 of Section A.1.

(b) Effect of number of groups and their placement on accuracy. Networks with the same structure are connected with the same type of line. Increasing group size drastically reduces BOPs with little impact on accuracy. The specific networks for each constellation are described in Table 6 of Section A.1.

Figure 2: Effect of (a) depth and (b) groups on accuracy as a function of BOPs on Imagenet. All results are reported for Stage I models. Best viewed in color.

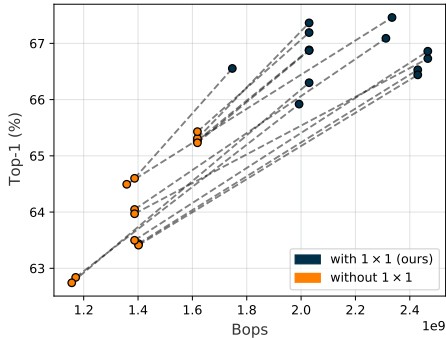

Figure 3: Effect of adding the $1 \times 1$ binary conv. layer after the grouped conv. layers. The dashed line connects same models with and without the $1 \times 1$ conv. layer.

Table 4: Comparison on ImageNet between a few structures explored. All methods have approximately the same number of FLOPS: $1.1 \times 10^8$. $N_i$: number of blocks in stage $i$, $E$: width expansion ratio, $G_i$: group size of convs at stage $i$. * - denotes model trained using AT+KD (Martinez et al., 2020).

| **Architecture** | **Acc. (%)** Stage II | | **Acc. (%)** Stage I | | **BOPS** |
|---|---|---|---|---|---|
| $N_0 N_1 N_2 N_3 - E - G_0 : G_1 : G_2 : G_3$ | **Top-1** | **Top-5** | **Top-1** | **Top-5** | $\times 10^9$ |
| 1242-2-4:4:16:32 | 66.3 | 86.5 | 68.4 | 87.7 | 1.3 |
| 1262-2-4:4:16:32 | 66.8 | 86.8 | 69.0 | 88.1 | 1.5 |
| 1282-2-4:4:16:32 | 67.7 | 87.4 | 69.7 | 88.8 | 1.7 |
| 1242-2-4:4:8:16 | 67.0 | 87.1 | 68.9 | 88.0 | 1.6 |
| 1262-2-4:4:8:16 | 67.6 | 87.3 | 69.7 | 88.6 | 1.9 |
| 1282-2-4:4:8:16 | 67.8 | 87.5 | 69.7 | 88.7 | 2.2 |
| 1262-2-4:8:8:16 | 67.5 | 87.5 | 69.5 | 88.6 | 1.7 |
| **1262-2-4:8:8:16\*** | **70.0** | **89.2** | 71.6 | 90.1 | 1.7 |

## 5    COMPARISON WITH STATE-OF-THE-ART

We compared our method against the current state-of-the-art in binary networks on the ImageNet dataset (Deng et al., 2009). Additional comparisons, including on CIFAR-100 (Krizhevsky et al., 2009), can be found in the supplementary material in Section A.2.

**Training:** The training procedure largely follows that of Martinez et al. (2020). In particular, we trained our networks using Adam optimizer (Kingma & Ba, 2014) for 75 epochs using a learning rate of $10^{-3}$ that is decreased by 10 at epoch $40, 55$ and 65. During Stage I, we set the weight decay to $10^{-5}$ and to 0 during Stage II. Furthermore, following Martinez et al. (2020), during the first 10 epochs, we apply a learning rate warm-up Goyal et al. (2017). The images are augmented following the common strategy used in prior-work (He et al., 2016) by randomly scaling and cropping the images to a resolution of $224 \times 224$px. In addition to this, to avoid overfitting of the given expert filters, we used Mixup (Zhang et al., 2017) with $\alpha = 0.2$. For testing, we followed the standard procedure of scaling the images to a resolution of 256px first and then center cropping them. All models were trained on 4 V100 GPUs and implemented using PyTorch (Paszke et al., 2019).

**Comparison against state-of-the-art:** Table 5 shows our results ImageNet. When compared against methods with similar capacity (Rastegari et al., 2016; Courbariaux et al., 2015; 2016; Bulat & Tzimiropoulos, 2019; Martinez et al., 2020) (bottom section of the table), our method improves on top of the currently best performing method of Martinez et al. (2020) by almost 6% in terms of top-1 accuracy. Furthermore, our approach surpasses the accuracy of significantly larger and slower networks (upper section) by a wide margin.

Finally, we compared our method against two very recent works that use NAS to search for binary networks. As the results from Table 5 (middle section) show, our method outperforms them, again by a large margin, while being significantly more efficient.

In terms of computational requirements, our method maintains the same overall budget, having an equal or slightly lower number of FLOPs and BOPs (see Table. 5). Although our method does increase the model size, by 2x for a model that uses 4 experts, the run-time memory largely remains the same. For additional details see Section A.5 in the supplementary material.

Table 5: Comparison with state-of-the-art binary models on ImageNet. The upper section includes models that increase the network size/capacity (last column shows the capacity scaling), while the middle one binary NAS methods. * - denotes models trained using AT+KD (Martinez et al., 2020). ‡ - denotes ours with an improved training scheme, see Section A.6 in supplementary material for details.

| Architecture | Accuracy (%) | | Operations | | # bits |
| --- | --- | --- | --- | --- | --- |
| | **Top-1** | **Top-5** | **BOPS** $\times 10^9$ | **FLOPS** $\times 10^8$ | (W/A) |
| ABC-Net ($M, N = 5$) (Lin et al., 2017) | 65.0 | 85.9 | 42.5 | 1.3 | $(1/1)\times 5^2$ |
| Struct. Approx. (Zhuang et al., 2019) | 66.3 | 86.6 | - | - | $(1/1)\times 4$ |
| CBCN (Liu et al., 2019) | 61.4 | 82.8 | - | - | $(1/1)\times 4$ |
| Ensemble (Zhu et al., 2019) | 61.0 | - | 10.6 | 7.8 | $(1/1)\times 6$ |
| BATS (Bulat et al., 2020) | 66.1 | 87.0 | 2.1 | 1.2 | (1/1) |
| BNAS-F (Kim et al., 2020) | 58.9 | 80.9 | 1.7 | 1.5 | (1/1) |
| BNAS-G (Kim et al., 2020) | 62.2 | 83.9 | 3.6 | 1.5 | (1/1) |
| BNN (Courbariaux et al., 2016) | 42.2 | 69.2 | 1.7 | 1.3 | 1/1 |
| XNOR-Net (Rastegari et al., 2016) | 51.2 | 73.2 | 1.7 | 1.3 | 1/1 |
| CCNN (Xu & Cheung, 2019) | 54.2 | 77.9 | 1.7 | 1.3 | 1/1 |
| Bi-Real Net (Liu et al., 2018) | 56.4 | 79.5 | 1.7 | 1.5 | 1/1 |
| Rethink. BNN (Helwegen et al., 2019) | 56.6 | 79.4 | 1.7 | 1.3 | 1/1 |
| XNOR-Net++ (Bulat & Tzimiropoulos, 2019) | 57.1 | 79.9 | 1.7 | 1.4 | 1/1 |
| IR-Net (Qin et al., 2020) | 58.1 | 80.0 | 1.7 | 1.3 | 1/1 |
| CI-Net (Wang et al., 2019) | 59.9 | 84.2 | - | - | 1/1 |
| Real-to-Bin* (Martinez et al., 2020) | 65.4 | 86.2 | 1.7 | 1.5 | 1/1 |
| **Ours** | **67.5** | **87.5** | **1.7** | **1.1** | 1/1 |
| **Ours*** | **70.0** | **89.2** | **1.7** | **1.1** | 1/1 |
| **Ours‡** | **71.2** | **90.1** | **1.7** | **1.1** | 1/1 |

**Comparison against CondConv:** As mentioned in Section 4.1, a direct application of Cond-Conv Yang et al. (2019) for the case of binary networks is problematic due to the so-called "double binarization problem", i.e. binarization of the weights and then of their linear combination is required. Herein, we verify this experimentally: when training a fully binarized network using Cond-Conv, we noticed a high degree of instability, especially during the initial phases of the training. For example, at the end of epoch 1, the accuracy of the binarized CondConv model is 1% vs 20% of the one using `EBConv`. The final accuracy of a binarized CondConv on Imagenet was 61.2% vs 63.8% compared to `EBConv`.

Additionally, as mentioned earlier, our proposed `EBConv` method uses less FLOPs (no multiplications required to combine the experts) and noticeably less memory at run-time (see Section A.5 in the appendix).

## 6 CONCLUSION

We proposed a three-fold approach for improving the accuracy of binary networks. Firstly, we improved model capacity at negligible cost. To this end, we proposed `EBConv`, the very first binary conditional computing layer which consists of data-specific expert binary filters and a very light-weight mechanism for selecting a single expert at a time. Secondly, we increased representation capacity by addressing the inherent information bottleneck in binary networks. For this purpose, we introduced an efficient width expansion mechanism which keeps the overall number of binary operations within the same budget. Thirdly, we improved network design, by proposing a principled binary network growth mechanism that unveils a set of network topologies of favorable properties. Overall, our method improves upon prior work, with no increase in computational cost by $\sim 6\%$, reaching a groundbreaking $\sim 71\%$ on ImageNet classification.

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

# A APPENDIX

## A.1 DETAILED NETWORK DEFINITIONS FOR FIG. 2

Table 6: Detailed network definitions for each of the constellations presented in Fig. 2. The location in each architecture is numbered from left to right (models with more BOPs are located towards the right).

(a) Network definitions for Fig. 2a

| Constell. # | Location | Configuration |
|---|---|---|
| 1 | 1 | 2222-1-1:1:1:1 |
|  | 2 | 1182-1-1:1:1:1 |
|  | 3 | 1282-1-1:1:1:1 |
|  | 4 | 1184-1-1:1:1:1 |
|  | 5 | 1188-1-1:1:1:1 |
|  | 6 | 1288-1-1:1:1:1 |
| 2 | 1 | 1132-2-4:4:4:4 |
|  | 2 | 1133-2-4:4:4:4 |
|  | 3 | 1233-2-4:4:4:4 |
|  | 4 | 2233-2-4:4:4:4 |
|  | 5 | 2282-2-4:4:4:4 |
|  | 6 | 1188-2-4:4:4:4 |
| 3 | 1 | 1242-2-4:4:8:16 |
|  | 2 | 1262-2-4:4:8:16 |
|  | 3 | 1282-2-4:4:8:16 |
|  | 4 | 1284-2-4:4:8:16 |

(b) Network definitions for Fig. 2b

| Constell. # | Location | Configuration |
|---|---|---|
| 1 | 1 | 1142-2-4:8:16:32 |
|  | 2 | 1142-2-4:4:16:16 |
|  | 3 | 1142-2-4:8:8:16 |
|  | 4 | 1142-2-4:4:8:16 |
| 2 | 1 | 1242-2-4:8:16:32 |
|  | 2 | 1242-2-4:4:16:16 |
|  | 3 | 1242-2-4:8:8:16 |
|  | 4 | 1242-2-4:4:8:16 |
| 3 | 1 | 1162-2-4:8:16:32 |
|  | 2 | 1162-2-4:4:16:16 |
|  | 3 | 1162-2-4:8:8:16 |
|  | 4 | 1162-2-4:4:8:16 |
| 4 | 2 | 1162-2-4:4:16:16 |
|  | 3 | 1162-2-4:8:8:16 |
|  | 4 | 1162-2-4:4:8:16 |

## A.2 ADDITIONAL COMPARISONS

Herein we present an extended comparison with both binary and low-bit quantization methods on ImageNet. As the results from Table 7 show, our method significantly surpasses both the binary and the more computationally expensive low-bit quantization networks. Similar results can be observed on the CIFAR-100 (Krizhevsky et al., 2009) dataset where our approach sets a new state-of-the-art result.

## A.3 ABLATION STUDIES

### A.3.1 REAL-VALUED DOWNSAMPLING DECOMPOSITION

The efficient width expansion mechanism of Section 4.2 preserves the amount of BOPs constant for binary convolutions. However, width expansion also affects the real-valued downsampling (linear) layers. To preserve the number of FLOPs constant, as width expands, for such a layer too, we propose to decompose it into two smaller ones so that the connection between them is reduced by a factor $r = k^2$, i.e. instead of using [Conv($C_{in}, C_{out}$)], we propose to use [Conv($C_{in}, \frac{C_{in}}{r}$) $\rightarrow$ Conv($\frac{C_{in}}{r}, C_{out}$)]. Herein, we explore a few variants by adding non-linearities between them. Our results, reported in Table 9, show that the non-linear versions are more expressive and bridge the gap caused by the decrease in the layer's size. The proposed adaption and the original one are depicted in Fig. 4.

Table 9: Comparison on ImageNet between various types of downsampling layers (Stage I models). All decompositions reduce the complexity by 4x.

| Decomposition | Accuracy (%) | |
|---|---|---|
|  | Top-1 | Top-5 |
| None | 67.4 | 87.2 |
| Linear | 66.5 | 86.6 |
| Non-linear (ReLU) | 67.2 | 87.2 |
| **Non-linear (PreLU)** | **67.5** | **87.3** |

Table 7: Comparison with state-of-the-art binary models on ImageNet, including against methods that use low-bit quantization (upper section) and ones that increase the network size/capacity (second section). The third section compares against binary NAS methods. Last column shows the capacity scaling used, while * - denotes models trained using AT+KD (Martinez et al., 2020). ‡ - denotes ours with an improved training scheme, see Section A.6.

| Architecture | Accuracy (%) | | Operations | | # bits |
|---|---|---|---|---|---|
| | Top-1 | Top-5 | BOPS $\times 10^9$ | FLOPS $\times 10^8$ | (W/A) |
| BWN (Courbariaux et al., 2016) | 60.8 | 83.0 | - | - | 1/32 |
| DSQ (Gong et al., 2019) | 63.7 | - | - | - | 1/32 |
| TTQ (Zhu et al., 2017) | 66.6 | 87.2 | - | - | 2/32 |
| QIL (Jung et al., 2019) | 65.7 | - | - | - | 2/2 |
| HWGQ (Cai et al., 2017) | 59.6 | 82.2 | - | - | 1/2 |
| LQ-Net (Zhang et al., 2018) | 59.6 | 82.2 | - | - | 1/2 |
| SYQ (Faraone et al., 2018) | 55.4 | 78.6 | - | - | 1/2 |
| DOREFA-Net (Zhou et al., 2016) | 62.6 | 84.4 | - | - | 1/2 |
| ABC-Net ($M, N = 1$) (Lin et al., 2017) | 42.2 | 67.6 | 1/1 | | |
| ABC-Net ($M, N = 5$) (Lin et al., 2017) | 65.0 | 85.9 | 42.5 | 1.3 | $(1/1) \times 5^2$ |
| Struct. Approx. (Zhuang et al., 2019) | 66.3 | 86.6 | - | - | $(1/1) \times 4$ |
| CBCN (Liu et al., 2019) | 61.4 | 82.8 | - | - | $(1/1) \times 4$ |
| Ensemble (Zhu et al., 2019) | 61.0 | - | 10.6 | 7.8 | $(1/1) \times 6$ |
| BATS (Bulat et al., 2020) | 66.1 | 87.0 | 2.1 | 1.2 | (1/1) |
| BNAS-F (Kim et al., 2020) | 58.9 | 80.9 | 1.7 | 1.5 | (1/1) |
| BNAS-G (Kim et al., 2020) | 62.2 | 83.9 | 3.6 | 1.5 | (1/1) |
| BNN (Courbariaux et al., 2016) | 42.2 | 69.2 | 1.7 | 1.3 | 1/1 |
| XNOR-Net (Rastegari et al., 2016) | 51.2 | 73.2 | 1.7 | 1.3 | 1/1 |
| CCNN Xu & Cheung (2019) | 54.2 | 77.9 | 1.7 | 1.3 | 1/1 |
| Bi-Real Net (Liu et al., 2018) | 56.4 | 79.5 | 1.7 | 1.5 | 1/1 |
| Rethink. BNN (Helwegen et al., 2019) | 56.6 | 79.4 | 1.7 | 1.3 | 1/1 |
| XNOR-Net++ (Bulat & Tzimiropoulos, 2019) | 57.1 | 79.9 | 1.7 | 1.4 | 1/1 |
| IR-Net (Qin et al., 2020) | 58.1 | 80.0 | 1.7 | 1.3 | 1/1 |
| CI-Net (Wang et al., 2019) | 59.9 | 84.2 | - | - | 1/1 |
| Real-to-Bin* (Martinez et al., 2020) | 65.4 | 86.2 | 1.7 | 1.5 | 1/1 |
| **Ours** | **67.5** | **87.5** | **1.7** | **1.1** | 1/1 |
| **Ours*** | **70.0** | **89.2** | **1.7** | **1.1** | 1/1 |
| **Ours‡** | **71.2** | **90.1** | **1.7** | **1.1** | 1/1 |

Table 8: Comparison with state-of-the-art binary models on CIFAR100. Last column shows the capacity scaling used, while * - denotes models trained using AT+KD (Martinez et al., 2020).

| Architecture | Accuracy (%) | # bits (W/A) |
|---|---|---|
| XNOR-Net (ResNet18) (Rastegari et al., 2016) | 66.1 | 1/1 |
| XNOR-Net (WRN40) (Rastegari et al., 2016) | 73.2 | 1/1 |
| CBCN (Liu et al., 2019) | 74.8 | $(1/1) \times 4$ |
| Real-to-Bin* (Martinez et al., 2020) | 76.2 | 1/1 |
| **Ours** | **76.5** | 1/1 |
| **Ours*** | **77.8** | 1/1 |

### A.3.2 DATA AUGMENTATION

Network binarization is considered to be an extreme case of regularization Courbariaux et al. (2015). However, recent work suggests that data augmentation remains an important, necessary aspect for successfully training accurate binary networks Martinez et al. (2020). Due to their lower representational power, Martinez et al. (2020) argues that, for the binarization stage, a weaker augmentation, compared to real-valued networks, should be used on large datasets such as ImageNet.

Table 10: Impact of temperature $\tau$ on accuracy (Stage I models) on ImageNet.

| $\tau$ | 0.02 | 1 | 5 | 25 |
|---|---|---|---|---|
| Top-1 acc. | 65.4 | **65.5** | 65.4 | 64.6 |

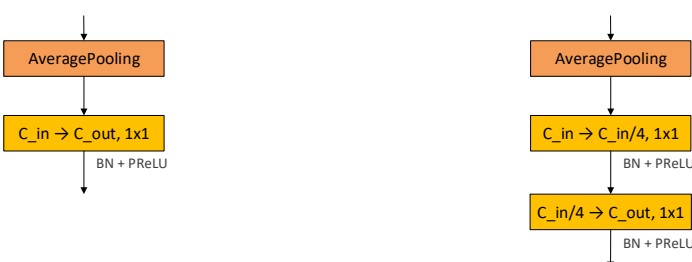

(a) The *vanilla* downsampling block.

(b) The proposed non-linear decomposition: We use 2 layers, with a non-linearity in-between, that maps $C_{in}$ to $C_{in}/n$ (here $n = 4$) and then back to $C_{out}$.

Figure 4: The (a) *vanilla* and (b) proposed downsampling block. This module is used in 3 places inside the network where the number of channels changes between macro-modules.

As opposed to this, we found that more aggressive augmentation, similar to the one used for real-valued networks in He et al. (2016) or mixup Zhang et al. (2017), leads to consistently better results. For example, using mixup on top of random scaling and cropping improves the results by 0.4%. In comparison, when we trained Real-to-Bin Martinez et al. (2020) with mixup, the accuracy dropped by 0.25% for Stage I, and 0.8% for Stage II. This suggests that, thanks to the proposed methods, we are getting closer than ever to the capacity of a real-valued model (which is amenable to stronger augmentations).

### A.3.3 EFFECT OF TEMPERATURE

One important component that influences the training efficiency of the gating mechanism is the softmax temperature $\tau$. As mentioned earlier, lower temperatures will produce spikier gradients while lower ones will induce the opposite. We explore the effect of various temperatures in Table 10. It can be seen that our results are stable over a wide range $\tau = [0.02, 5]$. Moreover, to validate the importance of using Eq. 4 for computing the gradients for back-propagation, we did an experiment where we replaced it with that of a sigmoid. Unsurprisingly, Stage I accuracy drops from 65.5% to 62.7%. This further highlights that the proposed form of the gating function is a key enabler for training higher performing models using `EBConv`.

### A.4 NETWORK ARCHITECTURE NAMING CONVENTION

This section clarifies the naming convention used in our paper: We define a network using the following notation $N_0 N_1 N_2 N_3 - E - G_0 : G_1 : G_2 : G_3$. Here $E$ is the expansion rate, defined as a multiplier with respect to a *vanilla* ResNet. For example a network with the first block having 128 output channels will have an expansion rate of 2. $N_i$ and $G_i$, $i = \{0, 1, 2, 3\}$ represent the number of convolutional blocks, and respectively, the number of groups used by all convolutions at each stage. Note that a ResNet has 4 stages. We graphically show the correspondence between this notation and the network structure in Fig. 5.

### A.5 MEMORY USAGE ANALYSIS

**Model storage size:** Current network binarization methods preserve the first and the last layer real-valued (Rastegari et al., 2016; Liu et al., 2018; Bulat & Tzimiropoulos, 2019). As such, for a ResNet-18 binary model trained on Imagenet, predicting 1000 classes, more than 2MB of the total space is taken by these parameters. As a result, our 4 expert model takes only 2x more space on a device. This is still noticeably less than binary models that attempt to increase their accuracy by increasing their model size (Lin et al., 2017) or by using an ensemble of binary networks (Zhu et al., 2019). Full results are shown in Table 11.

**Run-time memory:** In a typical deep network, the memory consumed by activations far outweigh that of the parameters (Jain et al., 2019). As such even a $\approx 4\times$ fold increase in the number of binary parameters (for the case of 4 experts) results in a small difference due to the above effect.

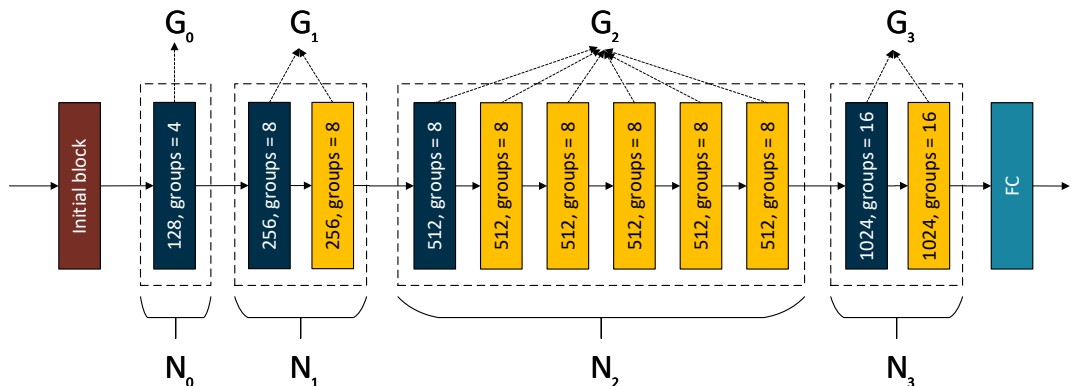

Figure 5: The overall network architecture of our final model defined as 1262-2-4:8:8:16. Inline with the current practice (Rastegari et al., 2016) the first (dark-red) and last layer (light-blue) are kept real. The yellow and dark-blue rectangles represent the binary residual blocks described in Sections 4.2 and 4.3 of the main paper with the text indicating the number of output channels and the number of groups. All blocks inside a macro-module, represented by a rectangle with dashed lines, operate at the same resolution, with the downsampling operation taking place at the first layer via strided convolution (dark-blue).

Table 11: Comparison with state-of-the-art binary models on ImageNet, including against methods that use low-bit quantization (upper section) and ones that increase the network size/capacity (second section). The third section compares against binary NAS methods. Last column shows the capacity scaling used, while * - denotes our model trained using AT+KD (Martinez et al., 2020). ‡ - denotes ours with an improved training scheme, see Section A.6.

| Architecture | Accuracy (%) | | Operations | | Model size | # bits |
|---|---|---|---|---|---|---|
| | Top-1 | Top-5 | BOPS $\times 10^9$ | FLOPS $\times 10^8$ | (MB) | (W/A) |
| ABC-Net ($M, N = 5$) (Lin et al., 2017) | 65.0 | 85.9 | 42.5 | 1.3 | 37.1 | $(1/1)\times 5^2$ |
| Struct. Approx. (Zhuang et al., 2019) | 66.3 | 86.6 | - | - | 7.7 | $(1/1)\times 4$ |
| Ensemble (Zhu et al., 2019) | 61.0 | - | 10.6 | 7.8 | 21 | $(1/1)\times 6$ |
| BNN (Courbariaux et al., 2016) | 42.2 | 69.2 | 1.7 | 1.3 | 3.5 | 1/1 |
| XNOR-Net (Rastegari et al., 2016) | 51.2 | 73.2 | 1.7 | 1.3 | 3.5 | 1/1 |
| Real-to-Bin (Martinez et al., 2020) | 65.4 | 86.2 | 1.7 | 1.5 | 4.0 | 1/1 |
| **Ours** (num. experts = 4) | **67.5** | **87.5** | **1.7** | **1.1** | 7.8 | 1/1 |
| **Ours\*** (num. experts = 4) | **70.0** | **89.2** | **1.7** | **1.1** | 7.8 | 1/1 |
| **Ours‡** (num. experts = 4) | **71.2** | **90.1** | **1.7** | **1.1** | 7.8 | 1/1 |

Furthermore, since only a single expert is active for a given input this effect is further reduced. This is confirmed by our measurements reported below. As a simple test bed for the later we leverage the built-in memory profiler from PyTorch: we measure and report the memory consumption for a convolutional layer with 512 input and output channels and a kernel size of $3 \times 3$. We set the input tensor to be of size $1 \times 512 \times 16 \times 16$. As it can be seen, since *a single expert* is active for a given image, our `EBConv` layer has a minimal impact on the memory usage. Bellow we show the profiler output with the operations sorted in descending order, based on memory. For brevity, we show only the top 10 contributors.

Memory profiling output for a normal convolutional layer, in descending order, based on memory:

```
---------------------    --------------    --------------
Name                     CPU Mem            Self CPU Mem
---------------------    --------------    --------------
conv2d                   512.00 Kb          0 b
convolution              512.00 Kb          0 b
_convolution             512.00 Kb          0 b
```

```
mkldnn_convolution          512.00 Kb          0 b
empty                       512.00 Kb          512.00 Kb
size                        0 b                0 b
contiguous                  0 b                0 b
as_strided_                 0 b                0 b
---------------------   --------------   ---------------
```

Memory profiling output for `EBConv` (ours), in descending order, based on memory:

```
---------------------   --------------   ---------------
Name                        CPU Mem          Self CPU Mem
---------------------   --------------   ---------------
empty                       514.02 Kb          514.02 Kb
conv2d                      512.00 Kb          0 b
convolution                 512.00 Kb          0 b
_convolution                512.00 Kb          0 b
mkldnn_convolution          512.00 Kb          0 b
adaptive_avg_pool2d         2.00 Kb            0 b
mean                        2.00 Kb            0 b
sum_out                     2.00 Kb            0 b
addmm                       16 b               16 b
softmax                     16 b               0 b
_softmax                    16 b               0 b
---------------------   --------------   ---------------
```

Memory profiling output for CondConv (Yang et al., 2019), in descending order, based on memory:

```
---------------------   --------------   ---------------
Name                        CPU Mem          Self CPU Mem
---------------------   --------------   ---------------
matmul                      9.00 Mb            0 b
mm                          9.00 Mb            0 b
resize_                     9.00 Mb            9.00 Mb
empty                       514.02 Kb          514.02 Kb
conv2d                      512.00 Kb          0 b
convolution                 512.00 Kb          0 b
_convolution                512.00 Kb          0 b
mkldnn_convolution          512.00 Kb          0 b
adaptive_avg_pool2d         2.00 Kb            0 b
mean                        2.00 Kb            0 b
```

Furthermore, as the profiler outputs show, for the case of CondConv (Yang et al., 2019), the additional multiplication operations required to combine the experts together significantly increase the run-time memory consumption, dominating it, in fact, for low batch sizes – a typical scenario for models deployed on mobile devices. This further showcases the efficiency of the proposed method. We note that the numbers of BOPs and FLOPs of our binary model will remain constant as the batch size increases because the number of operations itself does not change (with the exception of the linear increase induced by the number of samples within the batch). Additionally, for batch sizes larger than 1, there will be a small cost incurred for the actual reading (fetching) of the weights from the memory. However, this cost is insignificant. Finally, we note that, in most cases, when binary networks are deployed on edge devices, a batch size of 1 is expected.

## A.6 IMPROVED TRAINING SCHEME WITH STRONGER TEACHER

A key improvement proposed by Martinez et al. (2020) is the real-to-binary attention transfer and knowledge distillation mechanism. Therein, the authors suggest that using a stronger teacher does not improve the accuracy further, hence they use a real-valued ResNet-18 model as a teacher. Here, we speculate that the increase in representational capacity offered by the proposed model could benefit in fact from a stronger teacher. To validate this hypothesis, we train two real-valued teacher models of different capacity (controlled by depth): one scoring 72.5% Top-1 accuracy on ImageNet

and a larger one scoring 76.0%. As the results from Table 12 show, our model can exploit the knowledge contained in a stronger teacher network, improving the overall performance by 1.2%. Throughout the paper, we mark the results obtained using the stronger teacher with ‡.

We note that for training we largely preserve the gradual distillation approach described in (Martinez et al., 2020): In particular, at Step I, we train a full precision model with a structure that matches that of our binary network. At Step II, we use the previous model as a teacher and train a student with binary activations and real-valued weights. At the end of this step, we also perform our weight expansion strategy, propagating the trained weights across all experts following the optimization procedure described in Section 4.1. Finally, we use the model produced at the previous step as a teacher, training a fully binary network.

Table 12: Impact of the teacher used on the final accuracy of the model on ImageNet.

| FP32 Teacher | Binary Student |
|---|---|
| 72.5% | 70.0% |
| **76.0%** | **71.2%** |

## B  SUMMARY OF PRIOR WORK COMPONENTS USED

Herein we detail some of the methodological improvements proposed in prior works and also adopted for our strong baseline. We note, that most of these improvements are put together to create the strong baseline introduced in (Martinez et al., 2020) which is also the starting point of our work.

### B.1  PER-CHANNEL SCALING FACTORS

In order to minimize the reconstruction error between the full precision and binary convolution, in Rastegari et al. (2016), channel-wise real-valued scaling factors are used to modulate the output of the binary convolutions. In Rastegari et al. (2016), the authors proposed to calculate their values using an analytical solution that attempts to minimize the quantization error. The subsequent work of Bulat & Tzimiropoulos (2019) advocates for scaling factors learned via back-propagation by minimizing the task loss. In this work, we adopted the latter, learning one scaling factor per channel via back-propagation.

### B.2  DOUBLE-SKIP CONNECTIONS

Originally proposed by Liu et al. (2018), the double-skip connection mechanism adds a skip (i.e. an identity) connection around each binary convolutional layer. This is in contrast with a typical ResNet block (He et al., 2016) where the skip connection is applied at a block level. The main idea behind it is to preserve a real-valued signal alongside the binary one, improving overall the network's capacity. We also note that a network with skip connections around all binary layers will also preserve a full precision data path that can improve both the gradients and the information flow.

### B.3  PRELU ACTIVATIONS

Rastegari et al. (2016) showed that, despite the non-linear nature of binary networks, ReLU non-linearities added after the binary convolutions can further improve the model's accuracy. However, a ReLU completely eliminates negative values which in Bulat et al. (2019) is found to cause training instabilities. To alleviate this, Bulat et al. (2019) proposes to use a PReLU (He et al., 2015) activation instead. Thanks to its negative slope, it can better preserve the full spectrum of values produced by a binary convolution.

### B.4  2-STAGE BNN TRAINING

Binary neural networks are notably harder to optimize in comparison with their full precision counterparts (Rastegari et al., 2016; Courbariaux et al., 2015; 2016). Since most of the performance

degradation comes from binarizing the signal itself (*i.e.* activations), (Bulat et al., 2019) proposes a two-staged optimization strategy where the network is gradually binarized. During Stage I, a network with full precision weights and binary activations is trained. Then, in Stage II, a fully binary network is trained by initializing the model from the previous stage. As detailed in Section 5, the training scheduler in both stages is identical, with the exception of the weight decay, which for Stage II, is set to 0 Martinez et al. (2020).

### B.5 REAL-TO-BINARY KNOWLEDGE DISTILLATION

A reasonable objective for training highly accurate binary networks is that the features learned by a binary network should closely match those of a full precision one up to an approximation error induced by the quantization process (Rastegari et al., 2016; Martinez et al., 2020). In order to explicitly enforce this, Martinez et al. (2020) proposes to add after each block an $\ell_2$ loss between attention maps calculated from the binary and full precision activations. The full precision guiding signal typically comes from an identically structured pretrained real-valued model. To further enhance the efficacy of this process, Martinez et al. (2020) introduces a trainable data-driven scaling factor for modulating the output of the binary convolution. We note that this process is used only on top of our best models, and is marked in the tables using an "*".

Finally, we note that the gap between the real-valued and binary models is $\sim 3.5-4\%$ (depending on the configuration). In comparison, the next best method, Martinez et al. (2020) has a gap of $\sim 5\%$. This shows that while the proposed structure is tuned for binary networks, it will also perform well for the case of full precision networks. This is perhaps not too surprising since a model easy to binarize should be also easy to train in full precision, the opposite however is not always true.

## C    OVERALL BINARY NETWORKS STRUCTURE

Herein, we would like to add a few general notes about how a Binary Network is typically constructed. Following (Rastegari et al., 2016) and Courbariaux et al. (2016) most works binarize all convolutional layers except for the first and last ones (*i.e.* the classifier) alongside the batch normalization layers and the per-channel scaling factors (Rastegari et al., 2016; Lin et al., 2017; Liu et al., 2018; Liu et al., 2019; Zhu et al., 2019; Bulat & Tzimiropoulos, 2019; Qin et al., 2020; Wang et al., 2019; Martinez et al., 2020). Rastegari et al. (2016) notes that the first layer is not binarized because of the low number of channels (*i.e.* 3), the speed-up offered by binarization is not high. Furthermore, because the input to the network is typically real-valued, it is more natural to process it initially using real-valued operations. Similarly, the last layer sees smaller speedups in practice when binarized (Rastegari et al., 2016; Courbariaux et al., 2016) and often, depending on the task requires outputting continuous values instead of discrete ones. Finally, the batch normalization layers are kept real too since they significantly improve the training stability, and also implicitly adjust the quantization point.

We note that, as shown in Table 5, the vast majority of the operations are binary, with only a small proportion of them remaining real valued.

