# OpenReview forum: "High-Capacity Expert Binary Networks"
_ICLR.cc/2021/Conference — ICLR 2021 Poster_

### Official Review · AnonReviewer2 · 2020-10-18
**Official Blind Review #2**

**Rating:** 4
**Confidence:** 5

**Review:**

---Paper summary---:

This paper proposes three approaches to improve the performance of BNNs. 1) Training a super-network (ensemble of BNNs) and dynamically selecting one BNNs to execute conditioned on input. 2) Widening the layers with group convolution to enhance the representational capacity. 3)  Designing the architecture using EfficientNet considering the width, depth, groups and layer arrangement configurations simultaneously.  The effectiveness of the paper has been justified on ImageNet classification but can be further strengthened.

---Strength---:
+ The performance of the paper is promising. It will serve as a strong baseline for future works.
+ Using conditional dynamic routing to improve the BNNs capacity is interesting. Specifically, the paper proposes to learn a supernetwork (i.e., ensemble of several experts) during training and dynamically select the path during testing, which enhances the capacity while preserving the inference efficiency to some extent.

---Weaknesses---:

1: This paper ensembles some existing compression/NAS approaches to improve the performance of BNNs, which is not significant enough.
+ The dynamic routing strategy (conditional on input) has been widely explored. For example, the proposed dynamic formulation in this paper has been used in several studies [2, 3].
+ Varying width and depth has been extensively explored in the quantization literature, especially in AutoML based approaches [Shen et al. 2019, Bulat et al. 2020], to design high capacity quantized networks.
+ The effectiveness of the group convolution in BNNs was initially studied in [1]. Later works also incorporate the group convolution into the search space in NAS+BNNs methods [e.g., Bulat et al. 2020a] to reduce the complexity.

2:  In each layer, the paper introduces a full-precision fully-connected layer to decide which expert to use. However, for deeper networks, such as ResNet-101, it will include ~100 full-precision layers, which can be very expensive especially in BNNs. As a result, it deteriorates the benefits and practicability of the dynamic routing mechanism.

3:  The actual speedup, memory usage and energy consumption on edge devices (e.g., CPU/GPU/FPGA) or IoT devices must be reported. Even though the full-precision operations only account for a small amount of computations in statistics, it can have a big influence on the efficiency on platforms like FPGA.

4:  This paper proposes to learn the binary gates via gradient-based optimization while exploring the network structure via EfficientNet manner. Then the problem comes. This paper can formulate the <width, depth, groups and layer arrangement> as configuration vectors and optimize them using policy gradients and so on, with the binary gates learning unified in a gradient-based framework. So what is the advantage of the "semi-automated" method of EfficientNet over the gradient-based optimization? In addition, how about learning a policy agent via RL to predict the gates? I encourage the authors can add comparsions and discussions with these alternatives.

5: More experiments on deeper networks (e.g., ResNet-50) and other network structures (e.g., MobileNet) are needed to further strengthen the paper.


References:

[1]  MoBiNet: A Mobile Binary Network for Image Classification, in WACV 2020.

[2]  Dynamic Channel Pruning: Feature Boosting and Suppression, in ICLR2019.

[3]  Learning Dynamic Routing for Semantic Segmentation, in CVPR2020.

---

> ### Author Response · Authors · 2020-11-13
> **Response to reviewer 2 (part 1)**
>
> $\textbf{Q2.1}$: "This paper ensembles some existing compression/NAS approaches to improve the performance of BNNs, which is not significant enough."
>
> $\textbf{A2.1}$: We respectfully disagree with the reviewer, we provide concrete answers to all 3 points you raised below.
>
> $\textbf{Q2.2}$: "The dynamic routing strategy (conditional on input) has been widely explored. For example, the proposed dynamic formulation in this paper has been used in several studies [2, 3]."
>
> $\textbf{A2.2}$: What we claimed is that this is the very first work to explore conditional convolutions for binary networks. We never claimed that this is the first dynamic network in general. In fact, we dedicated the entire 2.2 section to this discussion.
> The only similarity of our work with [2] and [3] is that they use the general idea of dynamic computation. First of all, neither [2] or [3] are applied to binary networks. Secondly, compared to our work, both [2] and [3] are trying to solve different problems with different problem formulations, solutions and implementations. [2] is on pruning for dynamically selecting N channels from each layer -- our method selects a single expert (set of weights). Similarly, in [3], the authors attempt to dynamically define a network architecture, selecting, N routing direction paths for each active node.  Overall, the similarity is superficial, the domain, the goal and the implementation are in fact different.
>
> $\textbf{Q2.3}$: "Varying width and depth has been extensively explored in the quantization literature, especially in AutoML based approaches [Shen et al. 2019, Bulat et al. 2020], to design high capacity quantized networks."
>
> $\textbf{A2.3}$: We respectfully disagree with your comment: In Sections 4.2 and 4.3, we provide the first comprehensive study and analysis of the optimal values for the width, depth, layer distribution, block structure and number of groups for binary networks. The papers you mentioned do not provide such a study at all.
> Bulat et al. 2020 [4] doesn’t study the influence of width or depth. They also do not search for these dimensions. The authors simply report a few results with bigger models (by means of increasing the width), however they do not search nor draw any conclusions regarding this. Note that our method outperforms Bulat et al. 2020 [4] by ~6%.
> Similarly in [Shen et al. 2019] the authors simply search for the optimal width alone, selecting from a set of 6 possible values. We jointly search for the width, depth, layer arrangement and number of groups per block using also a completely different approach. Note that although [Shen et al. 2019] increases complexity (compared to our method) by 4-5 times, our method still outperforms it by ~1.5% (69.65% vs 71.2%).
>
> $\textbf{Q2.4}$: "The effectiveness of the group convolution in BNNs was initially studied in [1]. Later works also incorporate the group convolution into the search space in NAS+BNNs methods [e.g., Bulat et al. 2020a] to reduce the complexity."
>
> $\textbf{A2.4}$: We don’t claim using grouped convolutions as a novelty of the paper. Our method uses grouped convolutions in a novel way which is tailored to the nature of BNNs and is concretely supported by discussion and experimental evidence:
> * First of all, we don’t simply use grouped convs: in Section 4.2  we propose grouped convs combined with a width expansion strategy (which guarantees the same computational complexity) followed by aggregation based on 1x1 convolutions. This is proposed for the first time.
> * Secondly, this proposal is largely motivated by the Depth vs Width paragraph (Section 4.3, pp5) which discusses why and shows experimentally that increasing Width does not have the same impact as increasing Depth for BNNs (as opposed to real-valued networks) [7]. These are non-straightforward observations, proposals and results which (a) have not been discussed in prior work, (b) are shown for the first time to have such a large impact on BNN accuracy.
> * Thirdly,  this is the first work on binary networks to jointly study and search the following: the optimal group size per block, its width, the network depth, the layer arrangement.
>
> In [1] the authors, in an attempt to alleviate the issue of binarizing a MobileNet architecture, propose to mix the information and control the dependency between channels using a  K-dependency set, that as the authors put it: “has a flavor of group convolutions”. The authors report results for K={0,1,2,3}. They do not search for optimal number of groups. Their methodologies are completely different. Thank you for pointing [1], we will cite it.
> Similarly, Bulat et al. [4] does not study the influence of the group size, nor the influence of the depth or width. None of these parameters are part of their search. They simply introduce grouped convolutions, with a predefined, fixed, group size.

---

> > ### Comment · AnonReviewer2 · 2020-11-24
> > **Thanks for your response (part 1)**
> >
> > The authors’ feedback partially solved the novelty concern. Specifically, I need to clarify that dynamically selecting the experts for increasing the representational capability of BNNs is quite interesting. However, it raises the on-device efficiency concern as mentioned.
> >
> > Analyzing the Depth vs. Width is convincing and useful for the BNNs community, where I believe reducing the gradient path length can contribute to suppressing the quantization noise.
> >
> > However, jointly searching for the depth, width, group size and layer arrangement might be somewhat incremental considering the existing NAS or heuristic architecture design methods for BNNs (more references provided [S5, S6]). In addition, the search efficiency and effectiveness with the exponentially large search space by employing the EfficientNet manner might be challenged.
> >
> > References:
> >
> > [S5]: “Training wide residual networks for deployment using a single bit for each weight”, in ICLR2018.
> >
> > [S6]: “WRPN: Wide Reduced-Precision Networks”, in Arxiv 2017.

---

> > > ### Author Response · Authors · 2020-11-24
> > > **Response 2 to reviewer 2 (part 1)**
> > >
> > > We are glad that we managed to clarify some of the concerns you had. We hope that the additional replies provided will clarify all your remaining concerns. Thank you for your detailed feedback.
> > >
> > > $\textbf{Q2.15}$: "The authors’ feedback partially solved the novelty concern. Specifically, I need to clarify that dynamically selecting the experts for increasing the representational capability of BNNs is quite interesting. However, it raises the on-device efficiency concern as mentioned."
> > >
> > > $\textbf{A2.15}$: We thank the reviewer for acknowledging the novelty of the proposed EBConv layer. Regarding theoretical complexity, thank you for confirming the correctness of our theoretical analysis. Regarding on-device efficiency, please see our detailed response in A2.9.2 and A2.9.3
> > >
> > > $\textbf{Q2.16}$: "Analyzing the Depth vs. Width is convincing and useful for the BNNs community, where I believe reducing the gradient path length can contribute to suppressing the quantization noise."
> > >
> > > $\textbf{A2.16}$: We are glad that the reviewer found our proposed analysis useful.
> > >
> > > $\textbf{Q2.17}$: "However, jointly searching for the depth, width, group size and layer arrangement might be somewhat incremental considering the existing NAS or heuristic architecture design methods for BNNs (more references provided [S5, S6]). In addition, the search efficiency and effectiveness with the exponentially large search space by employing the EfficientNet manner might be challenged."
> > >
> > > $\textbf{A2.17}$: As you pointed out above, our search space defined by (depth, width, group size and layer arrangement) is ambitious, and we showed that our search strategy, which can be seen as a type of coordinate descent in that space, is far more effective in terms of identifying highly accurate and efficient binary architectures than other recent Binary NAS papers. We believe that this is an important finding in BNN literature.
> > >
> > > Furthermore, we do not see similarities between our work and  [S5,S6]. As the authors of [S5] state, their “main innovation is to introduce a simple fixed scaling method for each convolutional layer, that permits activations and gradients to flow through the network with minimum change in standard deviation” - so this seems to be a different subtopic. They simply use a WideResNet as a baseline -- and did not search for architectures at all. Furthermore they only binarize the weights, the activations are still in full precision. Furthermore, in S6 the authors simply use a 2x wider AlexNet model. There is no search space defined or network search at all. We believe that our work and [S5, S6] have very little in common.

---

> ### Author Response · Authors · 2020-11-13
> **Response to reviewer 2 (part 2)**
>
> $\textbf{Q2.5}$: "In each layer, the paper introduces a full-precision fully-connected layer to decide which expert to use. However, for deeper networks, such as ResNet-101, it will include ~100 full-precision layers, which can be very expensive especially in BNNs. As a result, it deteriorates the benefits and practicability of the dynamic routing mechanism."
>
> $\textbf{A2.5}$: This is incorrect - the cost is insignificant and as a proportion doesn’t grow as the network increases. The cost is also insignificant because the proposed layers are very small (num_channelsx4). For example, assuming a ResNet-101 the total cost of all these linear layers is below 500k FLOPs. To put it into perspective, a single batch norm layer, operating at 8x8 resolution and 2048 channels (such as the one found in a typical ResNet-101 in the last stage) will use more than 250k FLOPs.
> Almost all current state-of-the-art methods use real-valued ops in a form or another. The first and the last layer in all work we compare with are for example kept real alongside the batchnorms, the scaling factors (ex: XNOR-Net etc) and real-valued linear layers [6].
>
> $\textbf{Q2.6}$: "The actual speedup, memory usage and energy consumption on edge devices (e.g., CPU/GPU/FPGA) or IoT devices must be reported. Even though the full-precision operations only account for a small amount of computations in statistics, it can have a big influence on the efficiency on platforms like FPGA."
>
> $\textbf{A2.6}$: Our work proposes new ideas and methodologies for drastically improving the accuracy of BNNs. Measuring and creating efficient hardware implementations for example on CPU  goes beyond the scope of this paper. Likewise, we do not have the tools to implement, deploy and then accurately measure power consummation on IoT devices. Similarly, implementing neural networks on FPGA goes beyond the scope of this paper and requires a different type of technical expertise. At the time of writing, there is no support for binary convolutions on GPUs on any of the major available libraries. Moreover existing GPUs can rarely be considered the target platform for BNNs: they are optimized for vector multiplications, not for bitwise operations and are not present on many of low-powered edge devices.  Furthermore, we are not the only ones who omit such results: the large majority of binary neural networks papers we are comparing with do not report this kind of results due to the aforementioned technical expertise requirements which constitute a research field of its own.
>
> $\textbf{Q2.7}$: "This paper proposes to learn the binary gates via gradient-based optimization while exploring the network structure via EfficientNet manner. Then the problem comes. This paper can formulate the <width, depth, groups and layer arrangement> as configuration vectors and optimize them using policy gradients and so on, with the binary gates learning unified in a gradient-based framework. So what is the advantage of the "semi-automated" method of EfficientNet over the gradient-based optimization? In addition, how about learning a policy agent via RL to predict the gates? I encourage the authors can add comparsions and discussions with these alternatives."
>
> $\textbf{A2.7}$: The advantage of our semi-automated approach is clearly demonstrated by our strong results: we were able to report a groundbreaking 71% on ImageNet  largely improving upon the gradient-based optimization you suggested and deployed in recent Binary NAS ([4,5] published at ECCV20). As the results from Table 5 show, we significantly outperform both methods, by ~ 6% while being more efficient (in terms of BOPs and FLOPs).
> The advantages of our semi-automated method are multifold:
> * Our method is stable to train, in [4], Bulat et al, shows that searching in the binary domain is unstable without additional regularization tricks.
> * Applying [4,5] to large networks and data sets directly is computationally infeasible. Unlike [4,5], that perform the search on a proxy-dataset (CIFAR10) using a downsized network, we perform our search  directly on the target dataset (ImageNet) using the full sized model. This results in better architectures, more suitable for challenging tasks.
> * Our method produced networks that are significantly more accurate, at the same or lower computational cost.
>
> We didn’t find any RL based binary methods to compare with and training such models directly on the target datasets is prohibitive in terms of computational resources (thousands of GPU-hours are required to search such models on CIFAR10). We believe this direction on how to adapt RL based NAS to binary networks could be a potential research topic on its own.

---

> > ### Author Response · Authors · 2020-11-13
> > **Response to reviewer 2 (part 3)**
> >
> > $\textbf{Q2.8}$: "More experiments on deeper networks (e.g., ResNet-50) and other network structures (e.g., MobileNet) are needed to further strengthen the paper."
> >
> > $\textbf{A2.8}$:Thank you for your suggestion. However, we already performed experiments with a different number of layers in terms of depth. In general, while we derive our initial network from the ResNet18 structure, the subsequent proposed models diverge from it, using different number of blocks per stage, different width ratio between each stage etc. For example, the 2222 block arrangement described in our paper (corresponding to ResNet18 architecture) has 18 layers while the proposed 1282 arrangement without the extra 1x1 binary layers has 28 layers and the proposed 1282 arrangement with the extra 1x1 binary layers has 41 layers.
> > Finally, binarizing the MobileNet architecture was one of the first things that we tried in our research: unfortunately it didn’t work. It seems that this is also discussed in the reference you provide: [1].
> >
> > References:
> >
> > [1] MoBiNet: A Mobile Binary Network for Image Classification, in WACV 2020.
> >
> > [2] Dynamic Channel Pruning: Feature Boosting and Suppression, in ICLR2019.
> >
> > [3] Learning Dynamic Routing for Semantic Segmentation, in CVPR2020.
> >
> > [4] Adrian Bulat, Brais Martinez, and Georgios Tzimiropoulos. BATS: Binary architecture search, ECCV 2020
> >
> > [5] Kunal Pratap Singh, Dahyun Kim, and Jonghyun Choi. Learning architectures for binary networks, ECCV 2020
> >
> > [6] XNOR-Net: ImageNet Classification Using Binary Convolutional Neural Networks, Rategari et al, 2016
> >
> > [7] Wide Residual Networks, Zagoruyko&Komodakis,2016

---

> > > ### Comment · AnonReviewer2 · 2020-11-24
> > > **Thanks for your response (part 3)**
> > >
> > > I sincerely appreciate the authors’ quick and informative response. I raised this question mainly for two reasons:
> > >
> > > a):  Most of the BNNs literature report the accuracy on ResNet-50, so this paper should also test it for further comparison.
> > >
> > > b):  This paper ambitiously searches for the width, depth, layer distribution, block structure and number of groups for binary networks, resulting in an exponentially large search space, especially for deriving the initial network from ResNet-50. So I suspect that using the EfficientNet search manner might not be feasible enough in terms of the efficiency. Moreover, searching for MobileNet structure with depthwise convolutions can be more challenging.
> > >
> > > The main concerns still exist after reading the authors' feedback.

---

> > > > ### Author Response · Authors · 2020-11-24
> > > > **Response 2 to reviewer 2 (part 3)**
> > > >
> > > > $\textbf{Q2.12}$: "a): Most of the BNNs literature report the accuracy on ResNet-50, so this paper should also test it for further comparison."
> > > >
> > > > $\textbf{A2.12}$: Probably you have misunderstood this: most of the previous works on BNNs use primarily modifications of ResNet18, a few ResNet34 but *very few* ResNet50. To illustrate this let’s have a look at the state-of-the-art methods we compare with in Table 8:
> > > >
> > > > | Don't use ResNet50   |     Use ResNet50      |
> > > > |----------|:-------------:|
> > > > |Real-to-Bin* (Martinez et al., 2020)  |ABC-Net (M,N= 5) (Lin et al., 2017) |
> > > > |CI-Net (Wang et al., 2019) | Struct. Approx. (Zhuang et al., 2019) |
> > > > |IR-Net (Qin et al., 2020) | |
> > > > |XNOR-Net++ (Bulat & Tzimiropoulos, 2019) | |
> > > > |Rethink. BNN (Helwegen et al., 2019) | |
> > > > |Bi-Real Net (Liu et al., 2018) | |
> > > > |CCNN (Xu & Cheung, 2019) | |
> > > > |XNOR-Net (Rastegari et al., 2016) | |
> > > > |BNN (Courbariaux et al., 2016) | |
> > > > |Ensemble (Zhu et al., 2019) | |
> > > > |CBCN (Liu et al., 2019) | |
> > > > | BATS (Bulat et al., 2020) | |
> > > > | BNAS (Singh et al., 2020) | |
> > > >
> > > > For completeness below we show how our method compares against Struct. Approx. (Zhuang et al., 2019) and ABC-Net (M,N=5)  (Lin et al., 2017). Note, we select 2 representative models from the respective papers based on a ResNet-50 architecture:
> > > >
> > > > | Method   |      Top-1/Top-5      | BOPS |
> > > > |----------|:-------------:|-------------:|
> > > > |Ours | 71.2/90.1 | $1.7\times10^9$ |
> > > > |ABC-Net (M,N=5) (Lin et al., 2017) | 70.1/89.7 | ~$100\times10^9 $ (58x slower!) |
> > > > | Struct. Approx. (N=5) (Zhuang et al., 2019)| 69.5/89.2 | ~$20\times10^9$ (11x slower!) |
> > > >
> > > > Finally, we would like to point out again that our final model is not a ResNet but a newly found structure.
> > > >
> > > > $\textbf{Q2.13}$: "b): This paper ambitiously searches for the width, depth, layer distribution, block structure and number of groups for binary networks, resulting in an exponentially large search space, especially for deriving the initial network from ResNet-50. So I suspect that using the EfficientNet search manner might not be feasible enough in terms of the efficiency. Moreover, searching for MobileNet structure with depthwise convolutions can be more challenging."
> > > >
> > > > $\textbf{A2.13}$:  Thank you for recognising that our search space is ambitious -- we also hope that this is recognised as a strength in our paper. We do not claim that our approach is necessarily the only or even the best way to search this space. However, we showed that it can be effectively used to to find highly accurate and computationally efficient binary architectures which outperform recent binary NAS methods (BATS (Bulat et al., 2020) and BNAS-F (Singh et al., 2020)) by large margins.
> > > >
> > > > $\textbf{Q2.14}$:  "The main concerns still exist after reading the authors' feedback."
> > > >
> > > > $\textbf{A2.14}$: We hope that the above can clarify any remaining concerns.

---

> > ### Comment · AnonReviewer2 · 2020-11-24
> > **Thanks for your response (part 2)**
> >
> > I sincerely appreciate the authors’ quick and informative response. After reading the rebuttal, comments of other reviewers/AC and the revised manuscript, I raise my feedback as follows:
> >
> > 1:  In terms of Q2.5, the theoretical complexity analysis itself is correct. However, it might not stand in the real implementation scenario since each layer follows a sequential design. That means the binary branch should wait for the response of the floating-point branch for inference, where the latency might largely depend on the floating-point ops. Also the data exchange between the floating-point and binary arithmetic units can be quite expensive, with high energy and on-chip area cost.
> >
> > 2:  In terms of Q2.6, I’m not convinced by the response that the paper focuses on improving the accuracy of BNNs while the implementation on edge devices goes beyond the scope of the paper. My point of view is that the algorithm and implementation should be co-designed for real applicability. I acknowledge that many literature omit such results, but it does not mean the results should be omitted. Actually, some existing binary acceleration libraries are available [S1, S2, S3], and even the simulation is enough.
> >
> > 3: In terms of Q2.7, one can optimize the configurations directly by using sub-gradient methods, search gradients in natural evolution strategies [S4] in conjunction with other network parameters in a fully gradient-based optimization framework or using policy gradients in a RL framework. So it is still not clear to me about the motivation of using EfficientNet, since its search efficiency is not high enough.
> >
> > References:
> > [S1]:  “Bmxnet: An open-source binary neural network implementation based on mxnet”, in ACM MM 2017.
> >
> > [S2]:  “Finn: A framework for fast, scalable binarized neural network inference”, in FPGA 2017.
> >
> > [S3]:  “dabnn: A super fast inference framework for binary neural networks on arm devices”, in ACM MM 2019.
> >
> > [S4]: “ Stochastic search using the natural gradient”, in ICML 2009.

---

> > > ### Author Response · Authors · 2020-11-24
> > > **Response 2 to reviewer 2 (part 2)**
> > >
> > > $\textbf{Q2.9.1}$: "In terms of Q2.5, the theoretical complexity analysis itself is correct."
> > >
> > > $\textbf{A2.9.1}$: Thank you for confirming the correctness of our theoretical analysis.
> > >
> > > $\textbf{Q2.9.2}$: "However, it might not stand in the real implementation scenario since each layer follows a sequential design. That means the binary branch should wait for the response of the floating-point branch for inference, where the latency might largely depend on the floating-point ops."
> > >
> > > $\textbf{A2.9.2}$: The design can be parallelized: the gating function operates on the input before the batch-norm layer, and as such batch-norm operations and the gating function operations can be run in parallel. Furthermore both  batch-norm and gating function operations are very cheap (we would again like to point out that the additional operations for the selector function are insignificant as it is implemented using a small linear layer with up to 4 outputs.). To further emphasize how negligible the cost of this layer is, we measured, using pytorch, the runtime of a batch-norm layer and that of the gating function. Averaging the results over 5000 runs we found out that the linear selector is 200x(!) faster than the batch-norm layer. As an example, a batch-norm layer operating on an input of size 512x16x16 took 0.00354 sec. to execute, while the corresponding gating selector (512 channels in, 4 out) took 0.000013 sec.
> > >
> > > $\textbf{Q2.9.3}$:"Also the data exchange between the floating-point and binary arithmetic units can be quite expensive, with high energy and on-chip area cost."
> > >
> > > $\textbf{A2.9.3}$: We cannot argue against the above argument because an actual hardware implementation is required. However, the issue that you raised is related to the batch-norm operations which are *common to all Binary Networks*. Our method does *not* introduce any additional data exchange: there’s only expert selection. Only one single value needs to be communicated which is the selected expert. To summarize, you may be right with respect to the above (and as we admitted we are not hardware experts), but we believe that such issues exist in all BNNs, and our proposed design does not introduce new overheads.
> > >
> > > $\textbf{Q2.10}$:"In terms of Q2.6, I’m not convinced by the response that the paper focuses on improving the accuracy of BNNs while the implementation on edge devices goes beyond the scope of the paper. My point of view is that the algorithm and implementation should be co-designed for real applicability. I acknowledge that many literature omit such results, but it does not mean the results should be omitted. Actually, some existing binary acceleration libraries are available [S1, S2, S3], and even the simulation is enough."
> > >
> > > $\textbf{A2.10}$:  As already mentioned our contributions are on the methodological side in terms of training highly accurate binary networks. This is why we submitted our paper to ICLR. We also believe that our paper is not the first one to report Accuracy vs Flops (or Bops in our case). In fact, most published papers on efficient inference resort to this kind of metrics to report performance.
> > >
> > > As you also mentioned (almost) all binary network papers omit real-world latency results. The packages that you mentioned are good efforts but at a quite experimental level so we didn’t see a significant benefit in terms of running experiments there.
> > >
> > > We agree with you that the full impact of the work cannot be realised if there’s no proper hardware framework for evaluation. However sometimes methodological or theoretical results precede the actual hardware implementations and it’s only when these results are actually achieved that the hardware-related research communities devote more efforts into developing the corresponding hardware platforms.
> > >
> > > $\textbf{Q2.11}$:  "In terms of Q2.7, one can optimize the configurations directly by using sub-gradient methods, search gradients in natural evolution strategies [S4] in conjunction with other network parameters in a fully gradient-based optimization framework or using policy gradients in a RL framework. So it is still not clear to me about the motivation of using EfficientNet, since its search efficiency is not high enough."
> > >
> > > $\textbf{A2.11}$:  We do not claim that our approach is necessarily the only or even the best way to search this space. However, we showed that it can be effectively used to find highly accurate and computationally efficient binary architectures which outperform existing recent binary NAS methods (Bulat et al., 2020, Singh et al., 2020) by large margins. The methods you propose have not been previously used for searching binary network architectures, and hence their effectiveness is questionable. For example, (Bulat et al., 2020) showed that searching in the binary domain is unstable without additional regularization tricks. The strong results offered by our method further reinforce the advantages of the proposed approach. See also our previous response in A2.7.

---

### Official Review · AnonReviewer4 · 2020-10-28
**Strong results, and some questions about the experiments**

**Rating:** 6
**Confidence:** 4

**Review:**

The authors improved the performance of BNN by adopting group convolution and data-driven expert binary networks which choose a weight group that takes part in inference. In addition, by searching for the network architecture under the condition of the same number of operations, they achieved SOTA accuracy on ImageNet dataset.

The results in the paper are strong. Especially, the SOTA accuracy (71.2%) of the proposed network which even outperforms the full-precision ResNet-18 is impressive and could be the significant result in BNN research. The adopted data-driven expert network which increases the number of parameters, but maintains that of parameters participating in inference is also interesting.

Below is my remaining concerns and questions about the experiments.

1. Let us assume the condition that the proposed BNN which has four experts processes many images (i.e. multiple batches). In worst case, expert weights which a few images (extremely four) select by gating function can be all different, which gives rise for processors to have to fetch 4 times more parameters (i.e. all trained parameters) compared to single image inference.  So, I concern if under the condition of using multiple batches there is the performance degradation in terms of inference speed or energy in hardware compared to other BNN models.
2. Unlikely other previous works, mixup is used even on ImageNet. According to [1], it is stated that using mixup on ImageNet slightly degrades the accuracy. If an accuracy of your model trained without mixup (and if possible, accuracy of other models like Bi-Real Net or Real-to-Bin trained with mixup) is provided, the proposed model performance will be more clearly shown under the same training condition.

Minor comment:

1. I could not clearly understand the message or meaning of Fig. 1b. Detailed explanation of this figure seems to be needed.
2. I think it will be more helpful for readers to understand Fig. 2 if the authors provide clearer information about it: the detailed network information for each constellation in Fig. 2a and for type of lines in Fig. 2b.
3. I have a question if the searched network architecture is just optimized to BNN or not. I think this can be shown by measuring the accuracy drop of your model compared to the same full-precision network architecture.
4. As mentioned earlier, the experimental result is impressive, so it seems that other researchers or related people might want to use your model, but the stated training process seems to be little bit complicated. Are you going to make your code public?

Reference

[1] Brais Martinez, et al. Training binary neural networks with real-to-binary convolutions. ICLR, 2020.

---

> ### Author Response · Authors · 2020-11-13
> **Response to reviewer 4 (part 1)**
>
> $\textbf{Q4.1}$: "Let us assume the condition that the proposed BNN which has four experts processes many images (i.e. multiple batches). In worst case, expert weights which a few images (extremely four) select by gating function can be all different, which gives rise for processors to have to fetch 4 times more parameters (i.e. all trained parameters) compared to single image inference. So, I concern if under the condition of using multiple batches there is the performance degradation in terms of inference speed or energy in hardware compared to other BNN models."
>
> $\textbf{A4.1}$: Thank you, this is a good point for discussion which we will include in the paper. The numbers of BOPs and FLOPs of our binary model will remain constant as the batch size increases since the number of operations itself doesn’t change. As we mention in our paper (see Appending A.4), memory requirements do change (4x for 4 experts); however what dominates memory usage is the memory consumed by activations rather that of the parameters[1]. Furthermore, we presume that the 4 binary experts would typically be already loaded into memory. However, you are right that for batch sizes larger than 1 there will be a small cost incurred for the actual reading (fetching) of the weights from the memory. However in our tests in pytorch the potential increase was insignificant.
> Finally, we note that, in practice, for edge devices where BNNs are meant to be deployed the typical batch size will be 1. We already reported in the supplementary material the case of batch size of 1 where we show that our method comes at little to no extra cost regarding memory consumption.
>
> $\textbf{Q4.2}$: "Unlikely other previous works, mixup is used even on ImageNet. According to Real-to-BIn, it is stated that using mixup on ImageNet slightly degrades the accuracy. If an accuracy of your model trained without mixup (and if possible, accuracy of other models like Bi-Real Net or Real-to-Bin trained with mixup) is provided, the proposed model performance will be more clearly shown under the same training condition."
>
> $\textbf{A4.2}$: We did our best to keep the comparisons as fair as possible. As we report in the supplementary material, section A2.2, mixup gives us a gain of 0.4% in terms of accuracy. Note that only our model benefits from mix-up so we believe comparisons are fair. For example we already tried Real-to-Bin with mixup and the performance dropped by 0.25% for step 1, and 0.8% for step 2. We will add this text to section A2.2. Thank you again for your suggestion!
>
> $\textbf{Q4.3}$: "I could not clearly understand the message or meaning of Fig. 1b. Detailed explanation of this figure seems to be needed."
>
> $\textbf{A4.3}$: In Fig. 1b we use t-SNE to compute the similarity between all the samples from the ImageNet validation set based on their features, projecting them on a 2D space. Points located closer to each other are more semantically and visually similar. We color each data point according to the expert selected by the last EBConv from our network. The figures show that our experts learn a preference for certain classes, or groups of classes from Imagenet (notice the multiple clusters that emerge in the figure). This suggests that our EBConv layers learn semantically meaningful representations of the data.
>
> $\textbf{Q4.4}$: "I think it will be more helpful for readers to understand Fig. 2 if the authors provide clearer information about it: the detailed network information for each constellation in Fig. 2a and for type of lines in Fig. 2b."
>
> $\textbf{A4.4}$: Thank you for your suggestion, we will provide for each constellation the structure of the network too in the supplementary material

---

> ### Author Response · Authors · 2020-11-13
> **Response to reviewer 4 (part 2)**
>
> $\textbf{Q4.5}$: "I have a question if the searched network architecture is just optimized to BNN or not. I think this can be shown by measuring the accuracy drop of your model compared to the same full-precision network architecture."
>
> $\textbf{A4.5}$: That’s a very good point which we will add in the paper! We do have the results for the real-valued models since we used them as teachers for the distillation process and the gap is around 3.5-4% (depending on the configuration). In comparison, the next best method, Real-to-Bin has a gap of approx. 5%. This shows that while the proposed structure is tuned for binary networks, it will also perform very well for the full precision (FP32) networks too. This is perhaps not too surprising since a model easy to binarize should be also easy to train with FP32, the opposite however is not always true.
>
> $\textbf{Q4.6}$: "As mentioned earlier, the experimental result is impressive, so it seems that other researchers or related people might want to use your model, but the stated training process seems to be little bit complicated. Are you going to make your code public?"
>
> $\textbf{A4.6}$: Definitely, we will release both the training code and the pretrained models.
>
> References:
>
> [1]  Checkmate:  Breaking the memory wall with optimal tensorre materialization, Jain et al, 2019
>
> [2] Wide Residual Networks, Zagoruyko&Komodakis,2016

---

### Official Review · AnonReviewer1 · 2020-10-28
**Achieving very good accuracy with BNNs**

**Rating:** 5
**Confidence:** 2

**Review:**

Summary:
The paper addresses the problem of filling the gap between the performance of binary and real-valued networks. The authors propose a series of procedures to improve the model and representation capacity of binary neural networks. Different binary-network architectures are obtained through a new network-growing approach and compared.

Strengths:
Research on BNNs has a pretty short history if compared with studies on real-valued NN. It is a good idea to start translating some of the main tools from the standard NN literature to the binary setup.
According to the authors' claim, the proposed model greatly outperforms other existing and well-known binary networks.

Weaknesses:
The majority of the tools proposed for boosting the performance of binary networks are not new and have been already used in standard NNs. A discussion of the technical challenges associated with applying such tools to the binary setup would help understand the main contributions of the paper.
The inclusion of real-valued experts seems to make the final network not completely binary and it is not clear whether the advantages of BNN (e.g. the gain on computational costs) are preserved.
The network design step mainly consists of rearranging a series of pre-defined building blocks. It is not well explained how the architecture space is searched and how to interpret the results in Figure 2.

Questions:
- is the cost of Conditional computing included in the total cost when the main results are claimed (e.g. in "Without increasing the computational budget of previous works, our method improves upon the state-of-the-art by 6%")? More generally, when does the fixed number of BOPs include any training step?
- does the expert selection of the proposed method work better in the binary case than in real-valued networks?
- are the weights in each `expert' binary or real? Is it fair to compare the obtained hybrid model with real-to-bin?
- has Grouped Convolution with a similar scaling factor been already used somewhere?
- what is the difference between the proposed gradient-approximation method and standard "Straight-Through-Estimator" (STE)?

-----------------------
I acknowledge that I read and appreciated the author's response (both parts). The authors' reply mainly answers my questions, especially regarding the difference between applying the proposed techniques to the real and binary setups.

I agree with all authors comments but would tend to confirm my overall score for two reasons:

the architecture search method is not simply a block rearranging but looks more like a heuristic approach than a clear methodological contribution
the proposed mixing of real and binary weights may preserve the advantages of fully binary networks but, again, makes less clear the net contribution of the paper from a more theoretical perspective
However, as I recognize that the paper contains significant experimental results, I would be happy to support acceptance if all other reviewers agree on that.

---

> ### Author Response · Authors · 2020-11-13
> **Response to reviewer 1 (part 1)**
>
> $\textbf{Q1.1}$: "The majority of the tools proposed for boosting the performance of binary networks are not new and have been already used in standard NNs. A discussion of the technical challenges associated with applying such tools to the binary setup would help understand the main contributions of the paper."
>
> $\textbf{A1.1}$: First of all, due to the nature of binarization, architectural or methodological changes that benefit full-precision NNs do not result in accuracy improvements for BNNs. Direct application of techniques that work well for real-valued networks on Binary Networks has very often been unsuccessful. As an example, from our experiments, binarizing the MobileNet architecture performs poorly  (while in [1] it is shown that it does not converge at all). There are a multitude of reasons for this: in brief, this is due to the nature of binarization, where all values are discrete and restricted to 2 states only. This severely limits both the representational power of BNNs and causes issues during training since the true gradients are uninformative and their approximations make training harder to converge.
>
> Although conditional convs (CondConv) were previously used for standard NN, we show  (Section 5, “Comparison against CondConv and in Section 2.2, second paragraph) that directly applying the CondConv approach to binary networks is unsuitable. This can be attributed to the the fact that CondConv results in a linear combination of binary weights which is non-binary (see also Section 2.2, pp2, last paragraph); hence it requires a second binarization which as we show that doesn’t work well in practice.
>
> We further note that selecting the Best Expert as in our work vs a Mixture of Experts (MoE) as in CondConv is a different idea also requiring a different formulation and implementation (the formulation of Section 4.1 is not required in CondConv which is in general simpler).
>
> Likewise, while grouped convs are also known, our method uses them in a novel way which is tailored to the nature of BNNs and is concretely supported by discussion and experimental evidence: First of all, we don’t simply use grouped convs: in Section 4.2  we propose grouped convs combined with a width expansion strategy (which guarantees the same computational complexity) followed by aggregation based on 1x1 convolutions. This is proposed for the first time.  Secondly, this proposal is largely motivated by the Depth vs Width paragraph (Section 4.3, pp5) which discusses why and shows experimentally that increasing Width does not have the same impact as increasing Depth for BNNs (as opposed to real-valued networks) [4]. Thirdy,  this is the first work on binary networks to jointly study and search the following: the optimal group size per block, its width, the network depth, the layer arrangement.  These are non-straightforward observations, proposals and results which (a) have not been discussed in prior work, (b) are shown for the first time to have such a large impact on BNN accuracy.
>
> $\textbf{Q1.2}$: “The inclusion of real-valued experts seems to make the final network not completely binary and it is not clear whether the advantages of BNN (e.g. the gain on computational costs) are preserved”
>
> $\textbf{A1.2}$: The advantages are fully preserved. Their added cost is less than 0.001% (in terms of FLOPS) of the total computing budget. In fact all top performing BNN methods reported in Table 5 do use small amounts of  real valued operations reflected in the “FLOPS” column from Table 5 that quantifies the amount of full precision operations.

---

> ### Author Response · Authors · 2020-11-13
> **Response to reviewer 1 (part 2)**
>
> $\textbf{Q1.3}$: "The network design step mainly consists of rearranging a series of pre-defined building blocks. It is not well explained how the architecture space is searched and how to interpret the results in Figure 2."
>
> $\textbf{A1.3}$: We are not just rearranging the blocks. The network search space consists of the following degrees of freedom: a) rearranging the blocks (as pointed out by you),  but also b) defining the depth of the model, c) defining the width at each stage, and finally d) selecting the optimal number of groups per each level.
> In order to search for the optimal configuration, we gradually search in each direction separately while keeping all the others fixed. Then we identify a set of promising searching directions which we then combine to train new candidates. We repeat this step one more time and then from the final population of candidates we select the best models shown in Table 4. Note that this procedure largely outperforms the 2 recently proposed Binary NAS (~6%) methods from ECCV 2020. Thank you for this, we will add this discussion in the paper.
>
> In Fig. 2, we plot the accuracy of a set of networks grouped together into constellations. Every constellation represents a set of networks for which we keep fixed all searched parameters, except for the one studied. For example in Fig. 2a, we plot 3 sets of different networks as we plot 3 constellations. For each constellation all network parameters are fixed but the depth. As we see increasing depth has diminishing returns. Note that for clarity we plotted only a subset of the actual search space that we believe is sufficient to show the overall trend. We will try to further improve this Figure to make it more clear.
>
> $\textbf{Q1.4}$: "Is the cost of Conditional computing included in the total cost when the main results are claimed (e.g. in "Without increasing the computational budget of previous works, our method improves upon the state-of-the-art by 6%")? More generally, when does the fixed number of BOPs include any training step?"
>
> $\textbf{A1.4}$: Yes, they are included. In fact the cost of the selectors represents only around  0.001% of the total computational budget. This is due to the fact that the selectors (the linear layers) are small (num_of_channels x 4; for 4 experts).  In all tables, as it is common in the literature, only the inference cost is included. As we can see in Table 5, our method significantly outperforms all prior work using the same number of BOPs and less FLOPs.
>
> $\textbf{Q1.5}$: "Does the expert selection of the proposed method work better in the binary case than in real-valued networks?"
>
> $\textbf{A1.5}$: We did not test the proposed EBConv on its own on real-valued networks. The reason is that we don’t see a benefit of doing so and we don’t expect that it will work better, as CondConv will work there with no problems.
>
> $\textbf{Q1.6}$: "Are the weights in each `expert' binary or real? Is it fair to compare the obtained hybrid model with real-to-bin?
> A1.6:Thank you for your question, all the (4) experts are binary."
>
> $\textbf{Q1.7}$: "Has Grouped Convolution with a similar scaling factor been already used somewhere?"
>
> $\textbf{A1.7}$: The scaling factors are applied channewise so they are used in the same way as in standard binary convolution, see for example[3]. Grouped binary convolutions were used, for example in [2]. However in all previous works, the number of groups were fixed to a predefined value. The impact of this number and the placement of the groups inside the model were not studied before.
> In our paper, we don’t simply propose grouped convs: in Section 4.2  we propose grouped convs combined with a width expansion strategy (which guarantees the same computational complexity) followed by aggregation based on 1x1 convolutions. This is proposed for the first time. Furthermore, this is the first work to jointly study and search  for binary networks: the optimal group size per block, its width, the network depth, the layer arrangement.
>
> $\textbf{Q1.8}$: "What is the difference between the proposed gradient-approximation method and standard "Straight-Through-Estimator" (STE)?"
>
> $\textbf{A1.8}$: The STE is used to train the binary weights. It is implemented using an identity function (i.e. the gradients are simply copied from the layer above). In some implementations a clip function is also used to bound the gradients.
> Our proposed gradient-approximation is used to train the expert selection mechanism. It uses the derivative of the softmax with a temperature factor to control its “sharpness”.
>
> References:
>
> [1] MoBiNet: A Mobile Binary Network for Image Classification, Phan et. al, 2020.
>
> [2] BATS: Binary ArchitecTure Search, Bulat et al., 2020
>
> [3] XNOR-Net++: Improved binary neural networks, Bulat&Tzimiropoulos, 2019
>
> [4] Wide Residual Networks, Zagoruyko&Komodakis,2016

---

### Official Review · AnonReviewer3 · 2020-10-30
**A collection of some useful techniques**

**Rating:** 7
**Confidence:** 3

**Review:**

This paper proposes some techniques to improve the accuracy of binary networks without adding much computational overhead. To improve model capacity, the author proposes mixture-of-experts convolution with a winner-takes-all gating mechanisms. To deal with the limited representation power of binary activations, the paper proposes utilizing group convolutions. The performance is further improved by careful selection of hyperparameters and improved training techniques.

Clarity: This paper is pretty clear. The methodology is well-motivated and the algorithms / experiments are described clearly.
Originality: To the best of my knowledge, the utilization of mixture-of-experts and group convolutions for binary neural networks is novel.
Significance: The propose techniques, despite being simple, achieves good performance. They can be a new baseline for future research.

Overall I think this is a good paper and should be accepted.

---

> ### Author Response · Authors · 2020-11-13
> **Response to reviewer 3**
>
> We thank the reviewer for recognizing the novelty of our approach and their importance of our results on research on Binary Networks. We plan to release the code to facilitate future research.

---

### Author Response · Authors · 2020-11-13
**Overall response**

We thank the Reviewers for their time and comments and we hope that we addressed all points raised. We would be also more than happy to come back with more clarifications should they be further requested by the Reviewers.

---

### Comment · Area_Chair1 · 2020-11-13
**Suggestions for Clarification**

I have some suggestions where the clarity can be improved in my view.

* Contributions

The abstract describe 3 contributions. (a) is clear and I think it is quite innovative to make it work for BNNs. However (b) "introducing an efficient width expansion mechanism" and (c) "a principled binary network growth mechanism" look the same to me. In particular (c) appears to be just exploring (although rather systematically) depth, width, groups, etc. I personally like it more than NAS / BATS since it gives a lot of interpretable and valuable evidence. That evidence alone makes an experimental contribution. However I do not see where is the 'principled growth mechanism' or a methodological contribution that seems to be claimed (also mentioned in conclusion). Exploring configurations that have the same computation complexity is a sound approach, but not itself a contribution in my opinion. In addition, from the submission text, it also seemed to me that 'using grouped convolutions' is claimed as the contribution (b). If you do not claim this but something on top, consider rephrasing.

* Binary Networks

As noted by reviewers, real-valued computations are used in the network (at a fraction of computation cost). But not only in expert gating functions. There seem exist full real-valued paths from the input all the way to the output with learnable parameters on the way. Please correct me if this is wrong. I believe this is worth clarifying. Since sign on the input to the next layer is invariant of the scaling that you add on the output of the preceding layer, the scaling parameters seem to be influencing exclusively the real-valued skip connections. Thus regardless of what the binary stuff is doing, there is also a real-valued path that can learn to some extend and use the binary outputs as extra features. If this is so, it is worth clarifying. It may be also unclear to call this design a 'binary network'.

* Reproducibility

Related to the above, it is actually rather hard to trace out what kind of skip connections is used. The work by  Martinez et al. 2020 which you cite mentions double skip connections, redirecting to some earlier work. Some other detail like learnable coefficients are described in several variants in the prior work (e.g. per channel or per layer coefficients), linear transforms on skip connections, etc.
Overall, the amount of ingredients, many of them heuristic in nature, have grown significantly.
Please consider adding an appendix section listing all the ingredients of the network / training receipt that you inherit from the prior work, specifying for each one precisely the details, the link to the direct source and maybe adding a small comment how do you understand its working mechanism. I believe many researchers who want to understand the winning recipe and follow up would be very grateful for this.

* Experts

Before (4) you write "to use the Softmax function for approximating the gradients [of] φ(.)". I believe in (4) you wanted to use Softmax instead of $\phi$ so that the expression gives a gradient of Softmax. Would it not be simpler to write  $\frac{\partial \phi}{\partial z} := \frac{\partial }{\partial z} {\rm Softmax}(z)$? If you can give a citation or justification for this rule please do so.  Please clarify that "backpropagating gradients for non-selected experts" applies only to the gradient of the gating function and not to $x$.

---

> ### Author Response · Authors · 2020-11-17
> **Response to AC**
>
> Thank you for your suggestions and the time taken. We will update the paper to include all these changes ASAP.
>
> * Response to bullet point 1:
>
> In our view (b) and (c) are not the same. The word “growth” meant “search” or “exploration”. We will fix this to avoid misunderstanding.
>
> Contribution (b) refers to: proposing grouped convs combined with the width expansion strategy (which guarantees the same computational complexity) followed by aggregation based on 1x1 convolutions. This proposal is largely motivated by the Depth vs Width paragraph (Section 4.3, pp5) which discusses why and shows experimentally that increasing Width does not have the same impact as increasing Depth for BNNs (as opposed to real-valued networks).
>
> Contribution (b) on its own does not yield a highly optimized in terms of accuracy and complexity network. To this end, we built upon (b) to systematically search for the optimal group size per block, its width, the network depth, the layer arrangement. Since we show that this a more effective alternative to NAS, as you also pointed out, we still consider it as a contribution on its own. If this is to be considered more of an experimental rather than methodological contribution, it’s also up to one’s subjective interpretation -- we of course understand your perspective.
>
>
> * Response to bullet point 2:
>
> You’re right, and actually on top of your observations we would like to add that the first conv layer, the linear classifier from the end of the network, the BN layers, and the scaling factors following the binary convolutions are all real. However the complexity is largely dominated by the binary convolutions and that’s why these networks are called binary. We will make sure to clarify along with the reproducibility point you made below. Finally, we note that all competing binary networks we are comparing with are all based on these ideas (originating from the XNOR-Net paper of 2016): they are not adopted only by us in our paper.
>
> * Response to bullet point 3 :
>
> Thank you for your suggestion, this is a great point. We will update the paper over ASAP. Double skip connections were proposed originally in Bi-Real Net (Liu et al., 2018). We will also release code to facilitate the reproducibility of all the details mentioned.
>
> * Response to bullet point 4:
>
> We agree that the suggested notation is easier to interpret and following your advice we will use it to update the paper. Indeed, eq. 4 is used only for computing the gradients for the selector gating function, we will update the text to reflect this, distinguishing it from the STE used for the binary weights and activations.

---

### Author Response · Authors · 2020-11-22
**Updated manuscript**

We thank again the AC and the reviewers  for their time and comments. We updated the manuscript to reflect all the suggestions made and to address the points raised.

---

### Comment · ~Dahyun_Kim1 · 2021-01-19
**Reference clarification for BNAS**

Hi, thank you for your work on advancing the capacity of binary networks!

One small comment I have is that the citation for the BNAS paper is done with the arXiv version when the paper has been accepted to ECCV2020.

Please consider changing the citation to the ECCV version as follows: Dahyun Kim,  Kunal Pratap Singh, and Jonghyun Choi. Learning architectures for binary networks. In ECCV, 2020

---

### Comment · ~Xue_Geng1 · 2021-10-26
**Teacher Network Architecture**

Hi,

The paper mentioned they use two different teachers, one has 72.5% acc and the other 76%. Can the authors share the detailed architecture of the teacher?

---

### Decision · Program_Chairs · 2021-01-07
**Final Decision**

**Decision:**

Accept (Poster)

**Comment:**

## Summary
The paper advances the state of the art in training binary neural networks coming out to first place on ImageNet with a controlled computation budget. While any paper making a new record on ImageNet would be a serious candidate for acceptance, it is positive that this one achieves the goal by putting at work the mechanism of conditional computation, innovative for binary networks, and studying in a systematic and clear way how the network width and configuration can be varied while maintaining the computation budged.

## Review Process and Decision

The paper was thoroughly discussed by reviewers from different aspects. Several weaker spots have been identified (see below and final reviews), but no critical issues that would indicate a necessity of a major revision. In the end, reviewers agreed on acceptance although in some cases they have decided to keep their original <=5 ranking to reflect the scientific value to them from a more global perspective.
I think it is an example of well done modeling and experimental work: the work is very clear, uses sound methods, the experimental results are systematic and give interpretable evidence, which is in my experience is rather exceptional for the overall very empirical binary NNs field. I estimate high interest because of the concept of conditional computation put to work here and because of making a new record on ImageNet.

## Details

* Computation Cost

If such networks are to be deployed in low-power devices, the computation cost might need to be estimated more accurately. An example of such estimation is the work by
Ding et al. (2019) Regularizing Activation Distribution for Training Binarized Deep Networks,
where the energy and area are estimated using information from a semiconductor process design kit.

There is indeed a number of floating point operations around the binary convolutions: first and last layers, experts, skip connections with scale factors and non-linearities. The latency and cost of these operations may not be negligible on target devices. In particular Ding et al. (2019)  argue that XNOR-Net architecture is 3 times more costly because of floating point scale factors.
However the paper does a fair job in comparing in operation counts, which is a good proxy for many devices. The floating point computations needed in various places can be indeed further reduced to lower bit width, the research on quantization techniques shows this is possible and orthogonal to the contribution.

* Novelty of grouped convolutions design and search

The work of Phan et al. (CVPR 2020): Binarizing MobileNet via Evolution-based Searching
also proposed to search for best grouped convolution under computation budget constraints (evolutionary search method).
Strict budget constraint and merging results from different groups are somewhat novel and the prior work can be objectively contemporaneous.

* Clarity

Clarity of the paper has been improved by the revision. One remaining mysticism is still about the gradient estimator for the experts. The paper states: "we wish to back-propagate gradients for the non-selected experts", "allows meaningful gradients to flow to all experts during training". The problem is that since $\varphi(z)$ is binary one-hot on the forward pass, the gradient of the scalar product with $\varphi$ in (2) results in that in the backward pass only the selected expert receives the training signal and by no means all of them. This is regardless of how the gradient is propagated through $\varphi$. Maybe something is missing? I hope the authors can clarify in the final version. I do not consider it as a serious flow since this training scheme is not claimed as a contribution in any case.

One more point on the clarity: The paper claims that using experts increases the network representation power / capacity. While this seems logical, and follows the preceding work in real-valued NNs, the paper could provide additional evidence in terms of training performance of these models. Since the teacher with 76% accuracy is used in the distillation, I assume the training never reaches 100% training accuracy in any of the settings. Does the training accuracy improves with experts? This would be a helpful evidence for further work.

* Search method

The paper was further criticized for that the manual search of the architecture is a step back from automated search methods (NAS, BATS). However these methods are themselves a relaxation of discrete choices (experts, if you like), that need to keep all possible configurations at the same time, which may be less stable and too costly for real architectures and datasets. The principles of gradient-based architecture search are not entirely clear and the resulting models coming out of these methods typically give no insights regarding good (intelligent) design choices. At present, the systematic exploration with analysis of tradeoffs conducted is seen to have advantages.